# Quantization Meets Reasoning: Exploring and Mitigating Degradation of Low-Bit LLMs in Mathematical Reasoning

## Abstract

Low-bit post-training quantization (PTQ) is a practical route to deploy reasoning-capable LLMs under tight memory and latency budgets, yet it can markedly impair mathematical reasoning (drops up to 69.81% in our harder settings). We address two deployment-critical questions with process-level precision: **Where** along a step-structured solution does degradation first arise? **How** to mitigate it while staying in the low-bit regime? Across most widely used on computationally constrained scenarios PTQ methods (AWQ, GPTQ, SmoothQuant), open-source model families (Qwen, LLaMA; 0.5–7B), and math reasoning related benchmarks (GSM8K, MATH, AIME), we perform format-aligned chain-of-thought with step-aligned attribution and uncover two robust regularities: (i) PTQ disproportionately elevates method and execution errors relative to high-level conceptual mistakes; and (ii) failures emerge *early*, with the first vulnerable step flipping and cascading to the final answer. These regularities suggest a general intervention principle: restore local token-level margins exactly at the earliest failure frontier. We instantiate this principle as a lightweight *measure→locate→restore* loop that operates directly on the quantized model: detect the first faulty step, construct our **"Silver Bullet"** datasets, and apply small-scale supervised/preference tuning. In our settings, as few as 332 curated examples and 3-5 minutes of compute on a single GPU recover 4-bit weight math reasoning toward the full-precision baseline while preserving PTQ efficiency. Our framework is quantizer- and architecture-agnostic within the evaluated regimes, and turns low-bit degradation from a global accuracy problem into a local, reproducible process intervention.

## 1 Introduction

Transformer-based large language models (LLMs) such as LLaMA (Grattafiori et al., 2024), GPT (Achiam et al., 2023), and Qwen (Yang et al., 2024) have demonstrated strong performance on complex reasoning tasks, including mathematical competitions (Maxwell-Jia, 2025), code generation (Chen et al., 2021), and logical inference (Pan et al., 2023). Yet attaining reliable accuracy on such tasks typically requires large parameter counts. The resulting inference latency and memory footprint make deploying full-precision, ultra-large models impractical in many resource-constrained scenarios. To balance resource use and accuracy, model compression has been extensively studied, including quantization (Yang et al., 2019; Rokh et al., 2023), knowledge distillation (Hinton et al., 2015; Gou et al., 2021), and pruning (Han et al., 2015). Among these, post-training quantization (PTQ) (Banner et al., 2019) lowers precision to reduce memory and improve throughput, especially on edge hardware. However, recent evidence indicates that low-bit operation (e.g., INT4) can substantially degrade mathematical reasoning (Feng et al., 2024; Liu et al., 2025). This raises two practical questions for deployment: **Where** does degradation emerge in the reasoning process, and **How** can it be mitigated while remaining in the low-bit regime?

We study these questions through a systematic exploration of PTQ on widely used open-source model families and benchmarks. Concretely, we evaluate AWQ, GPTQ, and SmoothQuant on Qwen2.5 and LLaMA-3 across GSM8K (Cobbe et al., 2021), MATH (Hendrycks et al., 2021), and AIME (Maxwell-Jia, 2025). Using format-aligned chain-of-thought and *step-aligned* attribution, we characterize quantization-induced failures across model scales and task difficulty. Two patterns are

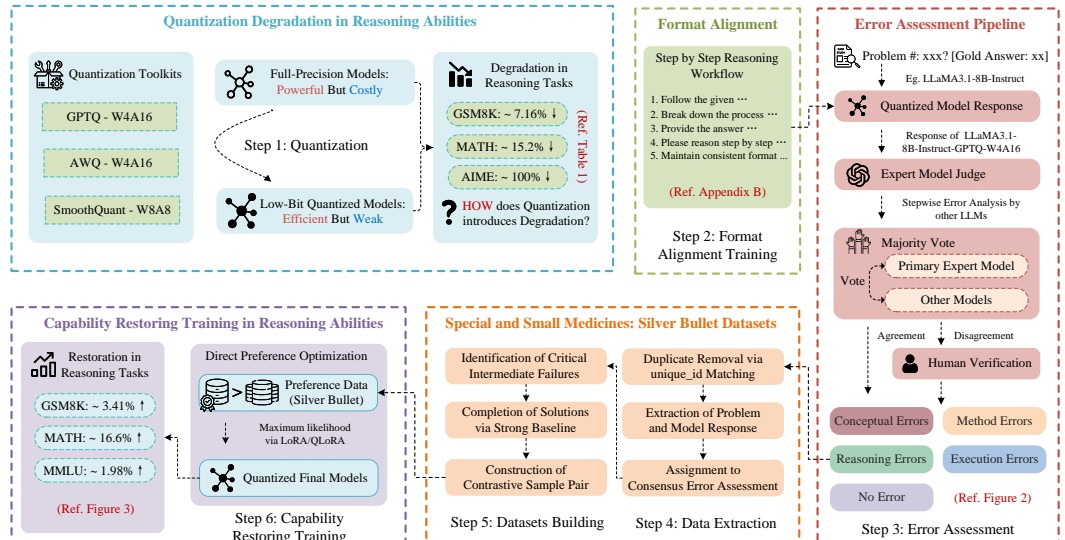

Figure 1: Pipeline of our study for investigating and restoring mathematical reasoning capabilities in quantized language models. We begin by identifying performance degradation caused by quantization, then apply format alignment training and a structured error assessment pipeline involving expert model judgments. Through this process, we analyze reasoning failures in step-by-step outputs. Targeted "Silver Bullet" datasets are constructed based on consensus error types, and used in DPO training to recover reasoning performance while maintaining the efficiency of low-bit models.

consistent: (i) PTQ predominantly increases *method* and *execution* errors (e.g., algorithm choice, rule application, carry/borrow, division/rounding), rather than high-level conceptual mistakes; and (ii) errors tend to emerge *early*, with the first vulnerable step flipping and cascading to the final answer. This diagnosis turns degradation into a targeted objective: restore token-level margins where collapse happens first.

Guided by this view, as shown on Figure 1, we adopt a lightweight *measure–locate–restore* loop that operates directly on the quantized model. We first locate the initial erroneous step, then apply small-scale supervised/preference tuning on a compact "Silver Bullet" set designed to target the observed weaknesses. In our experiments, fine-tuning on as few as 332 curated examples for 3–5 minutes on a single GPU is sufficient to recover the mathematical reasoning accuracy of W4A16 models toward their full-precision baselines, while preserving PTQ's memory and latency benefits.

We frame the study in the regime most relevant to practice: PTQ rather than quantization-aware training, so as to preserve the efficiency budget and expose unmodified low-bit failure modes. To cover current practice, we include AWQ, GPTQ, and SmoothQuant, which together span weight-only and weight–activation designs. Experiments use Qwen and LLaMA models at 0.5–7B—scales commonly deployed under constraints scenario and edge devices. Although our evidence is drawn from these choices, both the step-aligned measurement and the proposed *measure–locate–restore* loop intervention are architecture- and quantizer-agnostic by construction. Specifically, our primary contributions are as follows:

- We build a step-aligned measurement suite and hierarchical error taxonomy that expose a robust PTQ-induced shift toward *method* and *execution* errors, with earlier first-step flips, consistent across the most popular models, bit-widths, and benchmarks.
- We develop an automated chain-of-thought error–analysis pipeline (judge ensemble and light human audit) that attains 97.2% labeling accuracy on 9,908 failure cases, enabling fine-grained, reproducible attribution by error type and first faulty step.
- We introduce a compact *measure→locate→restore* loop that tunes the quantized model with targeted "Silver Bullet" pairs, recovering W4A16 mathematical reasoning to near full precision with just 332 curated examples and 3–5 minutes on a single GPU—without access to pretraining data.

## 2 RELATED WORKS

### 2.1 QUANTIZATION METHODS

Quantization is a computational efficiency optimization technique that maps high-precision tensors $X \in \mathbb{R}^{m \times n}$ into low-bit discrete representations. This work focuses on hardware-efficient uniform quantization, a linear mapping paradigm particularly suited for deployment on embedded systems with fixed-point arithmetic units. The method achieves significant reductions in model storage requirements and inference energy consumption while maintaining computational tractability.

For $b$-bit quantization, the mathematical formulation is expressed as:

$$\hat{X} = Q(X; b) = s \cdot \Pi_{\Omega(b)} \left( \frac{X}{s} \right) \tag{1}$$

where the quantization step size $s = \frac{\max(X) - \min(X)}{2^b - 1}$ dynamically adapts to the input distribution, effectively compressing the continuous floating-point space into an integer set $\Omega(b) = \{0, 1, \ldots, 2^b - 1\}$. The projection function $\Pi(\cdot)$ discretizes normalized values through nearest-neighbor rounding, with the rounding error being a primary source of quantization-induced precision loss. Notably, the step size $s$ governs the resolution of quantization intervals—larger dynamic ranges may sacrifice fine-grained details, necessitating calibration strategies for optimal parameter selection in practical implementations.

The engineering trade-offs of quantization manifest in multiple dimensions:

- **Bit-width Flexibility**: While aggressive 4-bit quantization reduces model size to 1/8 of its original footprint, it risks substantial accuracy degradation. Conversely, 8-bit quantization typically achieves near-full-precision performance in most scenarios.
- **Dynamic vs. Static Modes**: Dynamic quantization computes step sizes at runtime to adapt to input variations, whereas static quantization pre-calibrates parameters offline to minimize inference overhead.
- **Weight-only vs. Weight-activation**: Weight-only quantization restricts low-bit representation to model parameters, preserving activation precision for tasks sensitive to numerical stability. In contrast, weight-activation quantization jointly compresses both weights and intermediate activations, achieving higher memory efficiency at the cost of error accumulation.

Our methodology encompasses two complementary quantization approaches: (1) Post-training weight-only compression via AWQ (Lin et al., 2024) and GPTQ (Frantar et al., 2022), achieving 4-bit precision preservation through adaptive rounding strategies; (2) The SmoothQuant (Xiao et al., 2023) framework for joint weight-activation quantization, maintaining 8-bit numerical stability via learned scale migration. This dual-strategy design addresses distinct precision requirements: aggressive weight compression for memory efficiency versus moderate activation quantization for computational robustness. Comprehensive implementation protocols, including gradient-aware quantization grid adaptation and layer-wise sensitivity analysis, are detailed in Appendix A.

### 2.2 REASONING ABILITY OPTIMIZATION IN LARGE LANGUAGE MODELS

LLMs increasingly demonstrate strong general-purpose reasoning skills, spanning commonsense inference to domain-specific problem solving. Early evidence from Minerva (Lewkowycz et al., 2022) shows that scaling models and tailoring data can unlock advanced mathematical competence—one instance of the broader trend that rich intermediate computations boost reasoning fidelity. Prompt-engineering techniques such as Chain-of-Thought (Wei et al., 2022) and its code-generating variant Program-of-Thought (Chowdhery et al., 2023) further improve multi-step reasoning by encouraging models to decompose tasks into interpretable sub-steps.

Orthogonal to prompting, alignment research pursues systematic post-training refinements. Instruction tuning on diversified task mixtures (FLAN) (Wei et al., 2021) and lightweight data-curation pipelines (Alpaca) (Taori et al., 2023) make models broadly helpful, while Direct Preference Optimization (DPO) (Rafailov et al., 2024) offers sample-efficient preference learning without full RLHF. Reliability has been pushed along two complementary axes: self-consistency voting for answer

selection (Wang et al., 2022) and process-level supervision with stepwise reward models (Lightman et al., 2023b), both grounded in verifiable-reasoning theory (Creswell & Shanahan, 2022).

Building on these insights, we adopt process-supervised fine-tuning that obliges the model to articulate and justify each intermediate step. This explicit trace makes it possible to localize—and later ameliorate—reasoning failures introduced by low-bit quantization, providing a principled path toward efficient yet reliable LLM deployment.

## 3 METHODOLOGY

### 3.1 QUANTIZATION-INDUCED DEGRADATION: MEASUREMENT AND ATTRIBUTION

In this section, we investigate how low-bit quantization influences the reasoning performance of LLMs. Distinct from prior works, we examine each model's step-by-step solution trajectory and conduct a fine-grained quantitative–qualitative error analysis to pinpoint the root causes of reasoning failures. Our study centers on mathematically oriented tasks, which serve as a rigorous and representative proxy for general reasoning ability.

#### 3.1.1 QUANTIZATION

We conduct a comprehensive investigation into the effects of quantization techniques, encompasses two complementary quantization approaches: (1) Post-training weight-only compression via AWQ (Lin et al., 2024) and GPTQ (Frantar et al., 2022), achieving 4-bit weight precision preservation through adaptive rounding strategies and keep the data format of activations in 16-bit; (2) The SmoothQuant (Xiao et al., 2023) framework for joint weight-activation quantization, maintaining 8-bit numerical stability via learned scale migration. Through the systematic application of these most popular and wild-use quantization techniques, we provide a rigorous and balanced analysis of the resulting quantized models, offering valuable insights into their performance characteristics and trade-offs. Detailed algorithmic descriptions and mathematical derivations for all three methods are provided in Appendix A.

#### 3.1.2 FORMAT ALIGNMENT TRAINING

To address the challenge of inconsistent instruction following and irregular output formatting in model-generated solutions, we introduce a **format alignment stage**. This phase aims to instill in the model a structured, step-by-step reasoning workflow without altering its underlying mathematical knowledge. Crucially, the objective here is **NOT** to teach the model new mathematical facts or knowledge injection, but rather to ensure strict adherence to a prescribed output format, thereby enabling reliable qualitative and quantitative analysis of reasoning capability across quantized and full-precision variants.

We employ LoRA (Hu et al., 2021) and QLoRA (Dettmers et al., 2024) for full-precision model and quantized model respectively as lightweight adaptation techniques for format alignment. These methods efficiently align knowledge of step-by-step solution formats into the model's latent space without extensive retraining. This fine-tuning enables us to observe how multi-step reasoning is preserved or altered once the model is quantized, offering deeper insights into any capability loss induced by compression.

For alignment, we utilize the PRM800K dataset (Lightman et al., 2023a), which provides 800K step-level correctness annotations from 75K solutions to 12K problems. These annotations supply granular, step-by-step reasoning trajectories, equipping models to separate complex problem-solving processes into well-defined stages. To reinforce this structure, we adopt a consistent system prompt across training and evaluation, ensuring that the boundaries of logical steps and final answers are clearly delineated. This consistent, step-by-step alignment is a necessary foundation for our subsequent qualitative and quantitive analyses of quantization-induced degradation in mathematical reasoning. More details are presented on Appendix B

#### 3.1.3 DETAILED EXAMINATION OF REASONING PROCESS

**Qualitative Analysis.** To systematically investigate the underlying reasons for degradation in quantized models, we performed a qualitative error analysis inspired by established categorizations

Table 1: Comparison of quantization methods applied to the Llama-3 and Qwen2.5 model families. AWQ and GPTQ employ 4-bit weight and 16-bit activation quantization, whereas SQ (SmoothQuant) uses 8-bit weight and 8-bit activation quantization.

| | | GSM8K | | | | MATH | | | | AIME | | |
|---|---|---|---|---|---|---|---|---|---|---|---|---|
| | | Van. | AWQ (W4A16) | GPTQ (W4A16) | SQ (W8A8) | Van. | AWQ (W4A16) | GPTQ (W4A16) | SQ (W8A8) | Van. | AWQ (W4A16) | GPTQ (W4A16) | SQ (W8A8) |
| Llama | 3.1-8B-Inst. | 79.98 | 79.53 (0.56%) | 78.85 (1.41%) | 80.14 (-0.20%) | 50.72 | 46.44 (8.44%) | 46.04 (9.23%) | 50.26 (0.91%) | 10 | 1.11 (88.90%) | 5.56 (44.40%) | 6.67 (33.30%) |
| | 3.2-3B-Inst. | 77.26 | 74.45 (3.64%) | 71.57 (7.36%) | 77.71 (-0.58%) | 45.82 | 40.76 (11.04%) | 42.66 (6.90%) | 45.74 (0.17%) | 4.44 | 0 (100%) | 2.22 (50%) | 4.44 (0) |
| | 3.2-1B-Inst. | 45.56 | 39.12 (14.14%) | 39.58 (13.13%) | 45.19 (0.81%) | 20.58 | 16.2 (21.28%) | 15.78 (23.32%) | 20.96 (-1.85%) | 3.33 | 0 (100%) | 0 (100%) | 0 (100%) |
| Qwen2.5 | 7B-Inst. | 87.04 | 86.5 (0.62%) | 85.14 (2.18%) | 86.73 (0.36%) | 72.48 | 69.84 (3.64%) | 69.6 (3.97%) | 72.04 (0.61%) | 11.11 | 10 (9.99%) | 8.89 (19.98%) | 8.89 (19.98%) |
| | 3B-Inst. | 81.35 | 79.68 (2.05%) | 79.76 (1.95%) | 81.27 (0.1%) | 63.3 | 56 (11.53%) | 55.02 (13.08%) | 63.52 (-0.35%) | 4.44 | 3.33 (25%) | 3.33 (25%) | 2.22 (50%) |
| | 1.5B-Inst. | 68.23 | 61.11 (10.44%) | 59.89 (12.22%) | 68.46 (-0.34%) | 43.74 | 25.6 (41.47%) | 31.2 (28.67%) | 43.52 (0.5%) | 2.22 | 1.11 (50%) | 0 (100%) | 2.22 (0) |
| | 0.5B-Inst. | 43.37 | 27.07 (37.58%) | 26.61 (38.64%) | 41.55 (4.2%) | 23.98 | 8.02 (66.56%) | 7.24 (69.81%) | 24 (-0.08%) | 0 | 0 (0) | 0 (0) | 0 (0) |

*Notes: Van. denotes the vanilla full-precision baseline; Inst. is an abbreviation of Instruct.*

from previous literature (Brown et al., 2016), (Delastri & Lolang, 2023) and (Kurudirek et al., 2023), which categorize **real world student errors** in mathematical problem solving. Building on these frameworks, we conduct a qualitative analysis by classifying model-generated errors into seven fine-grained error types, organized under four high-level categories. The definitions of these error types are detailed as follows:

- **Conceptual Errors** arise when the model fundamentally misunderstands the underlying principles or context. This includes misgrasping core theories or overlooking domain-specific constraints (e.g., boundary conditions), leading to distorted problem framing and invalid solutions.
- **Method Errors** occur when mathematical methods are misapplied or chosen inappropriately. Typical cases include executing standard algorithms incorrectly, skipping key procedures, or misusing formulae in unsuitable contexts.
- **Execution Errors** stem from mistakes in arithmetic or symbolic manipulation, such as faulty calculations, erroneous expansions, or mislabeling variables. These slips compromise intermediate computations and ultimately the final answer.
- **Reasoning Errors** reflect flaws in logical flow, where inference steps do not follow coherently or essential links are omitted, creating gaps that render the conclusion unsupported.

**Quantitative Analysis and Error Assessment Pipeline.** To facilitate a rigorous and scalable evaluation of quantization-induced errors in reasoning tasks, we developed an automated assessment pipeline powered by state-of-the-art language models. This pipeline systematically processes model outputs and classifies errors according to our predefined *error_types_list* taxonomy. By leveraging a pre-trained transformer as the core evaluator, we reduce subjective bias and ensure consistent, reproducible error analyses across all experimental conditions. Furthermore, the computational scoring framework supports high-throughput performance assessment while preserving granularity in error categorization.

Our quantitative assessment pipeline comprises three primary stages:

**1. Expert Model Judgement**: For each instance in which a quantized model produces an incorrect answer, we employ a dedicated "expert model" to analyze the error. The expert model is tasked with: (a) identifying the first occurrence of an error, (b) specifying the exact step where the error is introduced, (c) assigning an error category based on a nested classification scheme, and (d) providing an explanation along with a confidence score for its determination.

**2. Majority Voting**: To curb hallucinations and improve evaluation reliability, we apply a three-stage majority-vote protocol to the outputs of five language models—*DeepSeek-R1* (Guo et al., 2025) (primary), GPT-4o, GPT-4, Qwen-Max (Yang et al., 2025), and DeepSeek-V3 (Liu et al., 2024). Instances of disagreement are flagged for further review, ensuring consistency and minimizing spurious judgments. **Rule1-Unanimous agreement**: If all four auxiliary models concur with the reference judgment from DeepSeek-R1, the answer is accepted. **Rule2-Simple majority**: If exactly

three auxiliary models concur with DeepSeek-R1, the answer is likewise accepted. **Rule3-Escalation**: Otherwise, the instance is forwarded to two independent human annotators for arbitration.

**3. Human Annotation**: For cases with conflicting assessments from the majority vote, we introduce two human annotators to manually review is conducted. The annotator need to follow the annotation document and review the explanations of five expert models then give the final assessment. Additionally, we also randomly sample 2% of the passed evaluated cases to verify the accuracy and consistency of the automated judgments. The annotation documents are detailed in Appendix C.

This pipeline is intentionally designed to be conservative and to avoid spurious "false consensus" in the automatic labels. Among several strong candidate judges, we select DeepSeek-R1 as the pivot because its reflective, stepwise chain-of-thought makes it particularly suitable for localizing the first erroneous step and producing structured explanations; in a small pilot on a random subset of failures, its error-type predictions also showed the highest agreement with two human annotators. For each incorrect model output, we then collect judgments from all five expert models and accept an automatic label only when at least three of the four auxiliary models concur with the pivot, that is, at least four out of five models agree on both the error type and its location; otherwise the instance is escalated to human annotators. In addition, we randomly sample two percent of the automatically accepted cases for manual audit.

Under this protocol, our automated error-assessment pipeline matches the final human judgment on 97.2% of 9,908 failure cases, with the remaining discrepancies concentrated in borderline situations, for example when the canonical answer is `"\\frac{11}{2}"` but the quantized model outputs the numerically equivalent `"5.5"` and some expert models still flag an error due to subtle reasoning or formatting differences. These observations suggest that our judge framework achieves high precision at the cost of slightly lower recall, which is appropriate for the downstream analyses in this paper; further implementation details and annotation guidelines are provided in Appendix C.

### 3.2 RESTORING REASONING ABILITIES IN QUANTIZED MODELS

#### 3.2.1 DATA EXTRACTION

Building on the analysis in Section 3.1.3, we construct our evaluation subset by filtering and categorizing problem instances according to model error types. First, to eliminate any risk of data leakage, we remove all overlapping examples between the MATH and MATH-500 test sets by matching on their `unique_id` fields. Next, for each quantized model, we identify those problems that the full-precision counterpart answers correctly but on which the quantized variant fails, based on the models' majority-vote outputs. We then collect the corresponding problem prompts and model-generated responses for these failure cases. Finally, leveraging the labels produced by our error-assessment pipeline, we assign each case to its consensus error category for downstream analysis.

#### 3.2.2 SILVER BULLET DATASETS BUILDING

During the execution of our error-assessment pipeline, we identify and record the exact reasoning step at which each quantized model initially commits an error. Our qualitative analysis indicates that many reasoning failures originate from incorrect intermediate computations or boundary adjustments, on which all subsequent solution steps heavily depend. Leveraging this observation, we construct a targeted counterexample dataset by truncating the incorrect reasoning traces precisely at the identified erroneous steps. Subsequently, we prompt powerful baseline models (Llama-3.2-70B and Qwen2.5-Max) to resume and complete these truncated solutions until the correct answers are derived. Consequently, we designate the original quantized models' erroneous partial solutions as negative samples, while adopting the accurately completed solutions generated by the larger models as positive samples. This approach yields our "Silver Bullet" datasets, specifically designed to facilitate downstream error correction and model fine-tuning.

#### 3.2.3 CAPABILITY RESTORING TRAINING

To reclaim the reasoning capability lost after low-bit quantization, we fine-tune each *quantized* model using **Direct Preference Optimization** (DPO) (Rafailov et al., 2024). Given a prompt $x$ and a pair of responses $(y^+, y^-)$ where $y^+$ is the correct answer and $y^-$ is the quantized model's incorrect

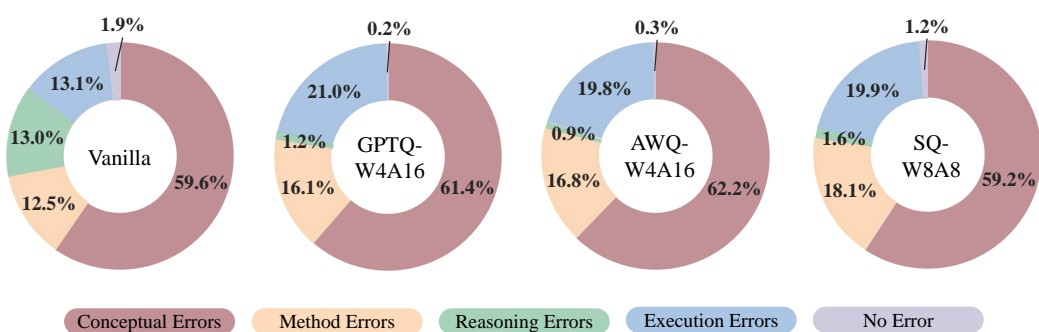

Figure 2: Error assessment results for full-precision and quantized models. For the full-precision model, we aggregate all problems it answered incorrectly; for each quantized model, we count only those problems that the full-precision model solved correctly but the quantized model failed, enabling comparison of quantization-induced changes across error dimensions.

answer, $y^+$ is prefered to $y^-$, DPO maximizes the log-likelihood gap between the two while softly constraining the new policy $\pi_\theta$ toward the frozen reference policy $\pi_{\text{ref}}$. The objective is

$$\mathcal{L}_{\text{DPO}}(\theta) = \mathbb{E}_{(x,y^+,y^-)\sim\mathcal{D}}\Big[\log\sigma\Big(\beta\big[\log\pi_\theta(y^+\,|\,x) - \log\pi_\theta(y^-\,|\,x)$$
$$- \big(\log\pi_{\text{ref}}(y^+\,|\,x) - \log\pi_{\text{ref}}(y^-\,|\,x)\big)\big]\Big)\Big]. \qquad (2)$$

where $\sigma$ is the sigmoid function and $\beta$ is an inverse-temperature hyper-parameter (we set $\beta = 1$). Because the reference gap is constant with respect to $\theta$, maximizing $\mathcal{L}_{\text{DPO}}$ is equivalent to minimizing $\text{KL}\big(\pi_\theta \,\|\, \pi_{\text{ref}}\big)$ subject to pairwise preference constraints, thus yielding a stable, RL-free preference-alignment procedure with solid theoretical footing.

We realize the adaptation using LoRA and 4-bit QLoRA. Across all experiments, we set the LoRA rank to 32 for every injected adapter matrix and optimize with a cosine learning-rate schedule (base learning rate $1 \times 10^{-6}$, warm-up ratio 0.1) under a global batch size of 8. Training minimizes the sigmoid preference loss implied by $\mathcal{L}_{\text{DPO}}$.

## 4 EXPERIMENTS

### 4.1 EVALUATING QUANTIZATION EFFECTS

In this phase of our study, we selected three benchmark datasets of varying difficulty levels to evaluate the degradation introduced by quantization across different reasoning complexities.

- **GSM8K** is a high-quality dataset of grade-school level math word problems released by OpenAI, containing 8,500 problems that typically require 2 to 8 steps of reasoning.
- **MATH** is a more challenging dataset composed of 12,500 competition-level high school math problems, covering seven mathematical domains including algebra, geometry, number theory, and probability and statistics, generally requires 15 or more steps of logical reasoning.
- **AIME** (American Invitational Mathematics Examination) is a high-difficulty International Mathematical Olympiad(IMO) competition designed for advanced middle and high school students with 90 problems (we combine problems from 2022-2025 for a widely evaluation).

We maintaining consistency in both the global batch size and the prompt with those used during alignment and evaluation. This setup ensures a fair comparison across all models. According to the Table 1 we find these two trends:

**Smaller-scale models suffer more severe losses in complex reasoning ability after quantization:** Across all quantization methods, smaller-scale models consistently demonstrate increased vulnerability to quantization. Specifically, the Qwen2.5-0.5B-Instruct model experiences accuracy drops exceeding 60% post-quantization, whereas the larger Qwen2.5-7B-Instruct model incurs only a

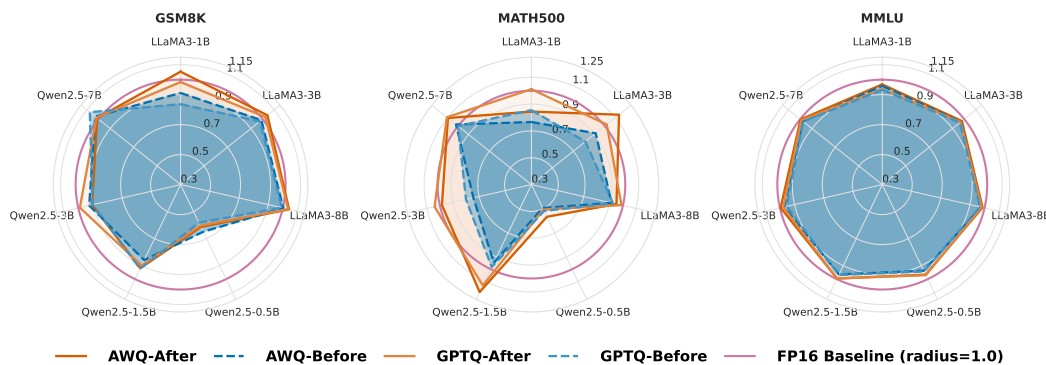

Figure 3: **Relative capability restoration with our method.** Radar values are normalized to each model's Vanilla-FP16 accuracy on the same benchmark (radius 1.0). Solid = After Restoration, dashed = Before Restoration (AWQ, GPTQ).

modest degradation of approximately 2–3%. This trend is also corroborated within the Llama3 model series. To rule out potential biases arising from larger models more readily fitting the calibration datasets, we further validated our findings using calibration datasets of varying sizes, consistently obtaining similar results. This evidence suggests that smaller models are more adversely affected by quantization-induced shifts in feature distributions, thereby experiencing more severe performance declines in complex mathematical reasoning tasks.

**Performance degradation becomes more pronounced as the task complexity increases:** We evaluated model accuracy across three mathematical reasoning benchmarks of varying difficulty levels. Our results indicate a clear trend wherein performance degradation exacerbates as task complexity rises. Among these, AIME represents the most challenging benchmark, with even full-precision models constrained by their scale unable to solve all problems effectively. The MATH dataset, characterized by evenly distributed difficulty tiers, poses intermediate-level complexity, while GSM8K is comparatively less challenging. Notably, quantized models exhibited relatively minor accuracy losses on the simpler GSM8K benchmark, with an average performance decline of only 7.16%. In contrast, the MATH dataset incurred a more pronounced average degradation of 15.18%. The most severe impact was observed on the highly challenging AIME benchmark, where quantization frequently led to complete failure in problem-solving capability.

We also evaluate the quantization-induced degradation on both thinking-mode models and larger-scale models for mathematical reasoning tasks, the detailed results are reported in Appendix E.1. Together with the corresponding evaluations under the same quantization settings on general-purpose benchmarks, the detailed results are reported in Appendix E.2.

### 4.2 ERROR TAXONOMY AND ITS SHIFT UNDER QUANTIZATION

**Error profile of full-precision models.** Using the assessment pipeline in Section 3.1.3, we examined every problem that the full-precision models answered incorrectly. *Conceptual Errors* were the most frequent (59.6%), while *Method*, *Execution*, and *Reasoning* Errors appeared at comparable rates of 12.5%, 13.0%, and 13.1%, respectively.

**Impact of quantization.** We next analyzed the subset of problems that full-precision models answered correctly but failed under quantized models. Across all three quantization methods, we observed a noticeable increase in the proportion of *Method Errors* and *Execution Errors*, suggesting that quantization predominantly impairs the model's ability to perform procedural operations and arithmetic execution (Feng et al., 2024). Supporting this observation, our case study reveals that quantized models exhibit greater difficulty in handling tasks involving basic arithmetic operations and numerical computation.

**Why Reasoning Errors seem to vanish.** The apparent reduction in Reasoning Errors after quantization arises from a statistical masking effect induced by our standardized evaluation protocol. Each trajectory is assigned a single error type according to the first erroneous step. In contrast, reasoning type failures in our taxonomy usually occur later in the solution when global logical consistency or boundary conditions are evaluated. Quantization increases the likelihood of earlier and simpler mistakes such as Conceptual or Execution Errors, and these early failures *"hide"* subsequent reasoning

Table 2: Ablation results on GSM8K, MATH500, and MMLU under AWQ/GPTQ (W4A16). Row labels denote training subsets: ALL-S = all error cases with step-aligned supervision from the first error; CE/ME/EE-S = only conceptual/method/execution errors with step alignment; Rand-NS = size-matched random sampling without step alignment; ALL-NS = all error cases without step alignment. Numbers are accuracy (%).

| | Llama-3.2-3B-Inst | | | | | | Qwen2.5-3B-Inst. | | | | | | Avg. |
|---|---|---|---|---|---|---|---|---|---|---|---|---|---|
| | AWQ(W4A16) | | | GPTQ(W4A16) | | | AWQ(W4A16) | | | GPTQ(W4A16) | | | |
| | GSM8K | MATH 500 | MMLU | GSM8K | MATH 500 | MMLU | GSM8K | MATH 500 | MMLU | GSM8K | MATH 500 | MMLU | |
| ALL-S | **74.3** | **36.8** | **60.57** | **73.01** | **33** | 59.9 | 68.84 | **38.4** | 64.8 | 75.21 | **40.6** | 63.63 | **57.42** |
| CE-S | 73.19 | 35.4 | 60.42 | 73.31 | 31.6 | 59.98 | 70.28 | 35.6 | 65 | 75.28 | 34.6 | 63.61 | **56.52** |
| ME-S | 73.62 | 32 | 60.45 | 72.4 | 30.4 | 59.86 | 69.6 | 34.8 | 65.01 | 76.27 | 32.6 | 63.81 | **55.90** |
| EE-S | 73.39 | 31.2 | 60.47 | 72.63 | 31.2 | 59.95 | 69.29 | 34.8 | 64.95 | 76.12 | 34 | 63.84 | **55.98** |
| Rand-NS | 73.84 | 30.9 | 60.51 | 72.71 | 31.2 | 59.97 | 70.05 | 32.5 | 64.92 | 74.13 | 34.4 | 63.71 | **55.73** |
| ALL-NS | 70.74 | 27.8 | 60.25 | 69.45 | 15 | 59.94 | 66.79 | 29.4 | 65.01 | 71.04 | 22.4 | 64.1 | **51.83** |

flaws from the statistics. Our case study confirms that many trajectories labeled as other error types still contain additional reasoning problems at later steps, although these later issues are not recorded because they occur after the first mistake. This masking effect therefore reflects how quantization reshapes the distribution of observed first errors under a reproducible and unambiguous protocol, not an actual disappearance of deeper reasoning mistakes. Absolute accuracies and examples are provided in Appendix D.

### 4.3 CAPABILITY RESTORATION

To prevent data leakage during evaluation, we report results on **MATH-500** (Lightman et al., 2023b), a 500-problem set that is disjoint from PRM800K yet mirrors the original MATH benchmark in topic coverage and difficulty. Performance on MATH-500 thus reflects genuine reasoning recovery rather than memorization. We also measure accuracy on GSM8K and MMLU (Hendrycks et al., 2020) to assess how well the restored model generalises to other reasoning-intensive tasks. The results are visually presented in Figure 3, with additional details provided in Appendix F.

### 4.4 ABLATION STUDY

To isolate the contributions of each component in our quantization recovery pipeline, we perform a series of ablation studies. Unless otherwise noted, all runs fix the training budget, optimizer, prompts, and decoding policy. We compare four variants of our training pairs (the *failure subset* refers to instances the quantized model answers incorrectly under baseline evaluation):

- ALL-STEP: all error cases from the failure subset; step-aligned supervision *resumes at the first-error step* (i.e., the first step where the model deviates from the gold solution) and continues step-wise along the gold trajectory to the final answer.
- CE/ME/EE-STEP: identical to ALL ERRORS-STEP but restricted to a single error type (*Conceptual-Error / Method-Error / Execution-Error*), with supervision resuming from the first-error step.
- RANDOM-NONSTEP: size-matched random sampling from the math corpus, independent of whether the quantized model fails; positives are the gold solutions, and *no* first-error resuming is applied.
- ALL-NONSTEP: all error cases from the failure subset, but *without* resuming from the first-error step; positives are the full gold solutions.

### 4.5 DISCUSSION

Synthesizing the results from Sections 4.3 and 4.4 together with the trends in Figure 2 and Table 2, we draw three main conclusions:

**(i) Targeted recovery with our "Silver Bullet" datasets.** Fine-tuning on the compact failure-targeted split substantially restores performance on MATH500 while also boosting GSM8K, and does so without hurting broad-domain reasoning as measured by MMLU. This intervention uses only a few hundred preference pairs and a few minutes of training on a single GPU, yet closes much of the gap between quantized and full-precision models. This confirms that the *"Silver Bullet"* provides *sample-efficient capability recovery rather than simple memorization*.

**(ii) Quantization disproportionately erodes procedural and executional skills.** Our error taxonomy shows that weight–activation quantization mainly increases *method* and *execution* errors—such as carrying out multi-step arithmetic or handling boundary conditions—rather than high-level conceptual reasoning. Because these mistakes often occur early, they propagate to invalidate otherwise correct derivations, explaining the steep drop on math-centric tasks.

**(iii) Step-wise positives outperform naive alternatives.** Ablation results in Table 2 is run under a strictly matched data and compute budget: each setting uses the same number of stepwise preference pairs and identical training hyperparameters, and only the selection of problems and traces is varied. Under this controlled setup, training on our error-targeted stepwise ALL split consistently outperforms both size-matched random supervision (RANDOM) and the non-stepwise variant (NON-STEP) that adopts full-precision derivations without restarting from the first erroneous step. On average, ALL improves accuracy by about 1.7 points over RANDOM and by about 5.6 points over NON-STEP, with the largest gap of 6.2 points on MATH500 for the most quantization-sensitive configuration (Qwen2.5-3B with GPTQ). These results indicate that locating the earliest quantization-induced failure and regenerating the remaining steps from that point provides a much stronger learning signal than unconditioned math supervision or full-solution positives, especially in the low-data regime we consider.

## 5 CONCLUSION

In this study, we present a systematic study of quantization-induced degradation in the mathematical reasoning abilities of large language models, revealing that low-bit post-training quantization especially harms smaller models' procedural and execution skills. To address this, we propose a lightweight recovery pipeline that combines step-aligned error analysis with targeted fine-tuning on compact, automatically constructed "Silver Bullet" datasets. Experiments show that, with minimal data and compute, quantized models can recover reasoning performance to match their full-precision counterparts while preserving efficiency and general capabilities. Our approach offers a practical and extensible solution for deploying quantized LLMs in resource-constrained settings, and opens avenues for robust reasoning restoration in broader domains.

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

# A   APPENDIX A

## A.1   AWQ

AWQ (Activation–Aware Weight Quantization) compensates for the long-tailed distribution of activations before the weight tensor is discretised. Let $\mathbf{A} \in \mathbb{R}^{B \times d}$ be the mini-batch activations and $\mathbf{W} \in \mathbb{R}^{d \times m}$ the corresponding weights. A positive scale vector $\boldsymbol{\gamma} \in \mathbb{R}_+^d$ is chosen such that

$$\widetilde{\mathbf{Y}} = \big(\mathbf{A} \odot \boldsymbol{\gamma}^{-1}\big)\big(\boldsymbol{\gamma} \odot Q(\mathbf{W})\big)^\top, \quad \gamma_k = \big(\text{mean} \, |\mathbf{A}_{,k}|^\alpha\big)\big(\text{mean} \, |\mathbf{W}_{k,}|^{-\beta}\big),$$

where $(\alpha, \beta) \in [0, 1]$ control the balance between activation and weight magnitudes and $Q(\cdot)$ denotes an asymmetric 4-bit quantiser. Because the rescaling is folded back into $\mathbf{W}$, the forward pass is identical to the unscaled INT4 kernel and incurs no extra latency.

## A.2   GPTQ

GPTQ formulates post-training quantisation as a blockwise least-squares problem over a small calibration set $\mathcal{C} = \{\mathbf{A}^{(i)}\}_{i=1}^{|\mathcal{C}|}$:

$$\widetilde{\mathbf{W}} = \arg \min_{\mathbf{W}' \in \mathcal{Q}} \sum_{i=1}^{|\mathcal{C}|} \big\|\mathbf{W}'\mathbf{A}^{(i)} - \mathbf{W}\mathbf{A}^{(i)}\big\|_F^2,$$

where $\mathcal{Q}$ is the set of weight tensors representable by the target bit-width. The optimisation proceeds greedily over 128-channel blocks. After quantising one block, GPTQ updates the remaining full-precision weights with a rank-$r$ approximation of the corresponding Hessian inverse, cheaply computed from second-order activation statistics. This strategy yields near-optimal INT4 weights with negligible calibration cost.

## A.3   SMOOTHQUANT

SmoothQuant jointly scales activations and weights so that both can be represented with the same uniform INT8 format. For each output channel, a learned scale $\sigma_k > 0$ migrates range from activations to weights:

$$\widetilde{\mathbf{Y}} = \big(\mathbf{A} \odot \sigma^{-1}\big)\big(Q(\mathbf{W} \odot \sigma)\big)^\top.$$

The scales $\{\sigma_k\}$ are obtained by minimising the worst-case per-channel quantisation error across the calibration set, typically using a few thousand tokens.[1] Once trained, the scales are fused into $\mathbf{W}$ and the model runs on standard INT8 kernels without auxiliary tensors or runtime re-scaling.

**Implementation Notes.** All three methods adopt per-channel affine quantisation. AWQ and GPTQ target 4-bit weights and retain FP16 activations, whereas SmoothQuant yields a fully INT8 model. We keep the original hyper-parameters recommended by the respective authors to ensure reproducibility across codebases.

---

[1] We use 2 048 tokens, following Xiao et al. (2023).

## B  PROMPT

---

**System Prompt**

Your task is to solving mathematical problems by following these rules:

1. Follow the given instructions strictly and ensure the output aligns with the expected format.

2. Break down the reasoning process into clear, logical steps using numbered lists (e.g., 1., 2., 3.) to sequentially present each step. Each reasoning step should be isolated and clearly described to enhance readability and logical flow.

3. After completing all reasoning steps, provide the final answer.

4. Please reason step by step, and put your final answer within `\boxed{}`. The final answer must be formatted as (`\boxed{your\_final\_answer}`). For example: Answer: `\boxed{42}`.

5. Maintain consistent formatting across training and evaluation phases to ensure the model outputs meet evaluation requirements seamlessly.

---

**Problem Statement (from PRM-800K Dataset Training Subsets)**

The product of the proper positive integer factors of $n$ can be written as $n^{(ax+b)/c}$, where $x$ is the number of positive divisors $n$ has, $c$ is a positive integer, and the greatest common factor of the three integers $a$, $b$, and $c$ is 1. What is $a + b + c$? **[Gold Answer: 1]**

---

**Response Sample(from PRM-800K Dataset Training Subsets)**

1. I notice that the problem involves the product of the proper positive integer factors of $n$, which means all the factors of $n$ except $n$ itself.
2. I also notice that the problem involves the number of positive divisors of $n$, which I'll call $x$.
3. I wonder if there is a way to relate the product of the proper factors of $n$ to $n$ and $x$.
4. I recall that the product of all the factors of $n$ is $n^{x/2}$, since each factor can be paired with its complement, and their product is $n$.
5. Therefore, the product of the proper factors of $n$ is $n^{x/2}/n$, which is $n^{(x-2)/2}$.
6. This means that the product of the proper factors of $n$ can be written as $n^{(ax+b)/c}$, where $a = 1$, $b = -2$, and $c = 2$.
7. The greatest common factor of $a$, $b$, and $c$ is 1, as required.
8. The answer is $a + b + c = 1 + (-2) + 2 = 1$. Answer: `\boxed{1}` **[Answer: 1]**

---

**Evaluation Prompt**

Role = 'HUMAN',
Prompt = "Your task is to solving mathematical problems by following these rules:

1. Follow the given instructions strictly and ensure the output aligns with the expected format.

2. Break down the reasoning process into clear, logical steps using numbered lists (e.g., 1., 2., 3.) to sequentially present each step. Each reasoning step should be isolated and clearly described to enhance readability and logical flow.

3. After completing all reasoning steps, provide the final answer.

4. Please reason step by step, and put your final answer within `\\boxed{}`. The final answer must be formatted as `\\boxed{your_final_answer}`. For example: Answer: `\\boxed{42}`.

5. Maintain consistent formatting across training and evaluation phases to ensure the model outputs meet evaluation requirements seamlessly.

Problem: `problem`"

---

---

**Assessment System Prompt**

**You are a helpful assistant that identifies and classifies errors in mathematical reasoning steps.**

**You will be given:**

- **Problem Statement:** A math problem statement.
- **Answers:** The right answer and answer from full-precision model and quantized model. Which model's answer is correct.
- **Full-Precision Reasoning:** The reasoning steps and final answer from a full-precision model.
- **Quantized-Model Reasoning:** The reasoning steps and final answer from a quantized model.
- **Error Type Definition:** The definition and explanation of error types.

**Your task:**

1. **Ground Truth Verification:** Compare both models' answers against the provided correct answer.

2. **Error Detection Protocol (Quantized Model):**
   If the quantized model is incorrect:

   1. Trace error origin using this hierarchy:
      - Conceptual Errors: `conceptual_misunderstanding`, `contextual_oversight`
      - Reasoning Errors: `logical_reasoning_error`
      - Method Errors: `procedural_error`, `formula_rule_error`
      - Execution Errors: `computational_error`, `symbolic_manipulation_error`
   2. Identify first point of divergence from correct reasoning.
   3. Classify using the most specific applicable type.
   4. Provide step-specific evidence.

3. **Conflict Resolution:**
   1. If multiple types apply, choose the earliest in the hierarchy.
   2. If ambiguity persists, use `procedural_error` as default.

**Return your analysis in the following JSON format strictly:**

```
{
  "quantized_error_analysis": {
    "primary_error_type": ["..."],
    "error_step"          : 1,
    "explanation"         : "Short evidence from reasoning steps",
    "confidence_score"    : 0.7  // between 0.7 and 1.0
  }
}
```

## C  HUMAN ANNOTATION GUIDEBOOK

PURPOSE

This guideline specifies the manual verification protocol applied to *disagreement cases* that survive the automated evaluation pipeline—namely the expert-LLM judges and the five-model majority vote. Annotators produce the *final ground-truth verdict* (error type, error step, explanation, confidence) for every instance in which

(a) the majority vote conflicts with the baseline judge **DeepSeek-R1**, or

(b) a "passed" case is randomly drawn for audit ($\approx 2\%$ of all cases).

MATERIALS PROVIDED

- `problem.txt`: problem statement.
- `answers.json`: correct answer, full-precision answer, quantized answer.
- `fp_trace.txt`, `qt_trace.txt`: step-by-step reasoning traces.
- `judge_outputs/`: five JSON files—DeepSeek-R1 (*baseline*), DeepSeek-V3, GPT-4o, GPT-4, Qwen-Max—each containing `primary_error_type`, `error_step`, `explanation`, `confidence_score`.
- `vote_summary.json`: ensemble result, per-model confidences, disagreement flag.

ERROR-TYPE TAXONOMY

1. **Conceptual Errors**: `conceptual_misunderstanding`, `contextual_oversight`

2. **Reasoning Errors**: `logical_reasoning_error`

3. **Method Errors**: `procedural_error`, `formula_rule_error`

4. **Execution Errors**: `computational_error`, `symbolic_manipulation_error`

*Earliest-precedence rule*: when multiple labels apply, choose the first that appears in the above list.

ANNOTATION PROCEDURE

1. **Answer verification**. Confirm which model(s) yield the correct final answer. If both are wrong, mark the case `dual_failure`.

2. **Locate first divergence**. Read `fp_trace` and `qt_trace` in parallel and find the earliest step where the quantized trace deviates from valid reasoning.

3. **Review automated evidence**. Inspect the five judge outputs and majority-vote result.

4. **Decision**.

   4.1. Adopt the ensemble consensus if at least three judges agree *unless* compelling counter-evidence exists.

   4.2. Otherwise, perform an independent assessment using the taxonomy in Sec. C.3.

5. **Label assignment**. Record `primary_error_type`, `error_step` (1-indexed), `explanation` ($\leq$ 40 words, quote the critical step), `confidence_score` (Sec. C.5).

6. **Quality flag**. Set `needs_second_opinion = true` if residual uncertainty remains.

CONFIDENCE-SCORE HEURISTIC

- **0.90 – 1.00**: clear evidence; $\geq$ 4 judges concur.
- **0.80 – 0.89**: moderate certainty; majority concurs; minor ambiguity.
- **0.70 – 0.79**: plausible but alternate interpretations exist; split vote (3–2 or worse).

OUTPUT SCHEMA

Annotators create `human_verdict.json` using

```
{
  "quantized_error_analysis": {
    "primary_error_type": "procedural_error",
    "error_step": 4,
    "explanation": "Applied quadratic formula with sign error at step 4.",
    "confidence_score": 0.83
  }
}
```

DIMENSION DEFINITION

- **Conceptual Errors** occur when the model exhibits a fundamental misunderstanding of the underlying principles or relevant context of the problem. This can manifest either as a conceptual misunderstanding, where the core ideas or foundational theories are not correctly grasped, resulting in an erroneous approach or framing of the problem; or as contextual oversight, in which crucial situational constraints or domain-specific factors (such as physical boundaries or geometric limitations) are overlooked, significantly distorting the solution process and its outcome.

- **Method Errors** refer to inaccuracies stemming from the improper selection or application of mathematical methods or established procedural approaches. Specifically, procedural errors happen when prescribed sequences or standard algorithms are incorrectly executed or entirely skipped, causing incomplete or invalid solutions. Formula rule errors are another subtype, characterized by the misuse or misapplication of relevant mathematical theorems, formulae, or rules—such as applying a formula in an inappropriate context—which fundamentally undermines the validity of the resulting calculations or conclusions.

- **Execution Errors** arise during the process of mathematical computation and symbolic manipulation. They encompass computational errors involving incorrect arithmetic or algebraic operations, such as flawed summations, erroneous expansions, or factorization mistakes, thus jeopardizing the accuracy of final answers. Additionally, symbolic manipulation errors include improper handling or representation of symbolic expressions, variables, or transformations. This could involve mislabeling variables or misinterpreting symbolic forms, leading to an incorrect representation and subsequent solution of the problem.

- **Reasoning Errors** involve flaws in the logical flow of problem-solving. Specifically, logical reasoning errors occur when there is a breakdown in the reasoning process itself, such that inference steps either do not logically follow one another or omit essential connections. This causes a logical gap or disconnect between the initial premises and the eventual conclusion, rendering the derived solution fundamentally flawed or unsupported.

DECISION AIDS

- *Conceptual misunderstanding*: misstates theorem before algebra begins.
- *Contextual oversight*: ignores domain restrictions or boundary conditions.
- *Logical reasoning error*: unsupported logical jump.
- *Procedural error*: applies an inappropriate solution method.
- *Formula rule error*: violates algebraic/derivative rule.
- *Computational error*: arithmetic slip.
- *Symbolic manipulation error*: incorrect simplification of an expression.

QUALITY CONTROL & ETHICS

- Two Annotators work independently; no discussion of live cases.
- Evaluate reasoning quality, not model identity; avoid bias.
- Flag any toxic or sensitive content present in traces.
- Project leads re-annotate 2 % of "agree" cases and all `needs_second_opinion` cases; overall agreement $< 95\%$ triggers targeted review.

**Note**.—DeepSeek-R1 is designated the baseline judge owing to its highest pilot agreement with human experts.

# D CASE STUDY

## D.1 EXAMPLE OF EXECUTION ERRORS

In Case #93, the task was to find the value of $c$ for which the circle defined by $x^2 - 10x + y^2 + 6y + c = 0$ has a radius of 1. The correct approach involves completing the square, resulting in $(x - 5)^2 + (y + 3)^2 = -c + 34$, and solving $-c + 34 = 1$ to find $c = 33$. However, the GPTQModel made a computational error by incorrectly rearranging the equation as $34 - c = 1$ leading to the wrong answer $c = 35$. This error stemmed from mishandling the sign of $c$ during algebraic manipulation, highlighting the importance of careful sign management in equation solving. The case #93 shows the **Execution Errors**.

---

**Problem #93**

For what value of $c$ will the circle with equation $x^2 - 10x + y^2 + 6y + c = 0$ have a radius of length 1?

**[Gold Answer: 33]**

---

**Response of `LLaMA-3.1-8B-Intruct-BF16` (Vanilla)**

1. To find the radius of a circle, I need to complete the square for both $x$ and $y$ terms in the equation.
2. I can rewrite the equation as $(x^2 - 10x) + (y^2 + 6y) = -c$.
3. To complete the square for $x^2 - 10x$, I add $(10/2)^2 = 25$ to both sides.
4. To complete the square for $y^2 + 6y$, I add $(6/2)^2 = 9$ to both sides.
5. This gives $(x^2 - 10x + 25) + (y^2 + 6y + 9) = -c + 25 + 9$.
6. I factor the left side as $(x - 5)^2 + (y + 3)^2$.
7. The right side simplifies to $-c + 34$.
8. Therefore the equation becomes $(x - 5)^2 + (y + 3)^2 = -c + 34$.
9. The radius of the circle is the square root of the right-hand side.
10. Setting the radius to 1 gives $-c + 34 = 1^2 = 1$.
11. Solving for $c$ yields $c = 34 - 1 = 33$.                          **[Answer: 33]**

---

**Response of `LLaMA-3.1-8B-Intruct-W4A16` (GPTQ Model)**

1. To find the radius, I first complete the square for the $x$ and $y$ terms and rewrite the equation in the form $(x - h)^2 + (y - k)^2 = r^2$, where $(h, k)$ is the centre and $r$ is the radius.
2. I begin with the $x$ terms, adding and subtracting the same value to form a perfect square.
3. Because the coefficient of $x$ is $-10$, I add and subtract $(-10/2)^2 = 25$ to obtain $x^2 - 10x + 25 - 25$.
4. Factoring gives $(x - 5)^2$ and the equation becomes $(x - 5)^2 - 25 + y^2 + 6y + c = 0$.
5. I repeat the process for the $y$ terms, adding and subtracting $(-6/2)^2 = 9$ to get $y^2 + 6y + 9 - 9$.
6. Factoring yields $(y + 3)^2$ and the equation is now $(x - 5)^2 - 25 + (y + 3)^2 - 9 + c = 0$.
7. Combining constants, I have $(x - 5)^2 + (y + 3)^2 - 34 + c = 0$.
8. Setting $r^2 = 1$ requires $-34 + c = 1$; solving gives $c = 35$.
9. Hence $c = 35$.                          **[Answer: 35]**

## D.2   EXAMPLE OF REASONING ERRORS

In Problem #128, the GPTQModel made a logical reasoning error when calculating the minimum number of miles Suzanne could walk in February. While the model correctly identified that February has 28 or 29 days, it missed the 27th day when calculating her walking schedule every third day, leading to an incorrect conclusion of 8 walking days and 32 miles. In reality, the correct number of walking days is 9 (3, 6, 9, 12, 15, 18, 21, 24, 27), resulting in a total of $9 \times 4 = 36$ miles. This error highlights the model's logical reasoning gap in iterating through sequential intervals accurately. The case #128 shows the **Reasoning Errors**.

---

### Problem #128

**Suzanne walks four miles every third day. What is the fewest number of miles she can walk in February?**

**Gold Answer: 36**

---

### Response of `LLaMA-3.1-8B-Intruct-BF16` (Vanilla)

1. To find the fewest number of miles Suzanne can walk in February, I need to figure out how many days she walks four miles in that month.

2. February has 28 or 29 days, depending on whether it is a leap year or not.

3. I do not know if this is a leap year, so I will assume it has 28 days.

4. If Suzanne walks four miles every third day, she walks on the 3rd, 6th, 9th, 12th, 15th, 18th, 21st, 24th, and 27th of February.

5. That is nine days, so she walks a total of $9 \times 4 = 36$ miles in February.

6. This is the fewest number of miles she can walk; if she skipped any of those days she would have to walk on a different day, increasing her total.

7. Therefore, the answer is 36 miles.                                    **[Answer: 36]**

---

### Response of `LLaMA-3.1-8B-Intruct-W4A16` (GPTQ Model)

1. To find the fewest number of miles Suzanne can walk in February, I must determine how many days she walks during that month.

2. February has 28 days, but some years have 29 days.

3. I will assume February has 28 days, since that is the most common case.

4. If Suzanne walks every third day, she will walk on days 3, 6, 9, 12, 15, 18, 21, and 24.

5. That is a total of 8 days, and she will walk 4 miles on each of those days.

6. Thus, the fewest number of miles she can walk in February is $8 \times 4 = 32$ miles.
   **[Answer: 32]**

### D.3 EXAMPLE OF NO ERROR

Despite the care taken in designing our extraction scripts, a small number of predictions remain hard to classify, leading to a residual "**No Errors**" category. A strong judge model can usually flag these edge cases; for instance, in case #3812, Deepseek-R1 correctly returns **No Errors** after a meticulous comparison. We subsequently review such instances and update the final labels accordingly.

Most ambiguities stem from multiple notations for the same numeric value—particularly decimals versus fractions and natural versus programming syntax—e.g., $1/2$, $0.5$, \frac{1}{2}, $5E-01$, and $5 \times 10^{-1}$.

---

**Problem #3812**

**What is the probability that when a fair 6-sided die is rolled, a prime number faces up?**
**Gold Answer:** $\frac{1}{2}$

---

**Response of** `LLaMA-3.1-8B-Intruct-W4A16` **(GPTQ Model)**

1. To find the probability of rolling a prime number on a fair 6-sided die, I need to know how many prime numbers appear on such a die and how many faces there are in total.

2. A prime number is a positive integer divisible only by 1 and itself. The prime numbers on a 6-sided die are 2, 3, and 5.

3. There are 3 prime numbers on a 6-sided die, so the probability of rolling a prime number is 3 out of 6, or $1/2$.

4. Converting this fraction to a decimal, I divide $1$ by $2$ to obtain $0.5$.

5. Alternatively, multiplying $1$ by $100$ and dividing by $2$ gives $50\%$.

6. Therefore, the probability of rolling a prime number on a fair 6-sided die is $0.5$ or $50\%$. **[Answer: 0.5]**

---

**Judgement of** `Deepseek-R1` **(Primary Expert Model)**

The quantized model's answer ($0.5$) is numerically equivalent to the correct answer ($1/2$). All reasoning steps are logically valid and mathematically correct, with no conceptual, procedural, or computational errors detected. **[No_Error: []]**

---

### D.4 EXAMPLE OF CONFLICTING JUDGMENTS

Case #342 illustrates our conflict-resolution protocol when the judge models disagree on an error label. We perform a five-way cross-model validation using Deepseek-R1, Deepseek-V3, ChatGPT, GPT-4o, and Qwen-Max. Each model independently assigns an error category to the quantized trace, and the final label is set by majority vote.

If the vote is inconclusive (e.g., a 2-2-1 split), a human annotator re-examines the example. The annotator consults (i) each model's confidence score, (ii) the accompanying explanations, and (iii) the step-by-step reasoning provided by Deepseek-R1, together with the raw model outputs. This double-check ensures that every ambiguous case receives a consistent, well-justified error type.

---

**Problem #342**

**What integer $x$ satisfies $\frac{1}{4} < \frac{x}{7} < \frac{1}{3}$?**
**Gold Answer: 2**

---

**Response of `LLaMA-3.1-8B-Intruct-W4A16` (GPTQ Model)**

1. To solve this inequality, I need to isolate $x$ on one side.
2. I can start by multiplying both sides by $4$, since $4$ is a common denominator for the fractions.
3. This gives me $1 < \frac{4x}{7} < \frac{4}{3}$.
4. Now I can multiply both sides by $7$, since $7$ is a common denominator for the fractions.
5. This gives me $7 < 4x < \frac{28}{3}$.
6. To make the inequality easier to read, I can divide both sides by $4$.
7. This gives me $\frac{7}{4} < x < \frac{7}{1}$.
8. I can simplify the fractions by writing them as mixed numbers or decimals.
9. This gives me $1.75 < x < 7$.
10. Since $x$ is an integer, I need to find the integers that satisfy this inequality.
11. The smallest integer greater than $1.75$ is $2$.
12. The largest integer less than $7$ is $6$.
13. Therefore, the integers that satisfy the inequality are $2, 3, 4, 5, 6$.        **[Answer: 2,3,4,5,6]**

**Judgement of `Deepseek-R1` (Primary Expert Model)**

When multiplying $4/3$ by $7$, the correct result is $28/3 \approx 9.33$. In step 5 the quantized model treats $7 \times (4/3)$ as $28/3$ but, in step 7, fails to divide this by $4$ correctly, giving an upper bound of $7$ instead of $7/3 \approx 2.33$. **[Execution_Errors: computational_error]**

**Judgement of `Deepseek-V3`**

The quantized model incorrectly multiplies both sides by $4$ instead of $7$, the proper common denominator. This leads to a procedural error that propagates through the subsequent steps. **[Method_Errors: procedural_error]**

**Judgement of ChatGPT**

The model simplifies the inequality to $\frac{7}{4} < x < \frac{7}{1}$, overlooking that the upper limit should be $\left(\frac{28}{3}\right) \div 4 = \frac{7}{3}$, not $\frac{7}{1}$. **[Conceptual_Errors: contextual_oversight]**

**Judgement of `GPT-4O`**

The quantized model reports the range $\frac{7}{4} < x < \frac{7}{1}$ instead of the correct $\frac{7}{4} < x < \frac{7}{3}$, yielding an incorrect set of integer solutions. **[Method_Errors: procedural_error]**

**Judgement of `Qwen-Max`**

The model simplifies $\frac{7}{4} < x < \frac{28}{3}$ to $1.75 < x < 7$. The upper bound $\frac{28}{3}$ is approximately $9.33$, not $7$. **[Conceptual_Errors: conceptual_misunderstanding]**

# E   EXPERIMENT RESULTS

## E.1   QUANTIZATION ON LARGER MODELS

To strengthen the generality of our conclusions and bringing potential insights, we have conducted additional experiments on the Qwen3 series, which natively support a "thinking mode" as the default configuration. These models already integrate internal CoT-like mechanisms, making them representative of both standard and "thinking" (CoT) LLMs. We evaluated Qwen3-8B, Qwen3-14B, and Qwen3-32B under identical quantization settings (bit-width, calibration set, and hyperparameters) as other qutization methods.

Table 3: Performance of Qwen3 models on MATH under different quantization methods.

| | Qwen3-8B | | | Qwen3-14B | | | Qwen3-32B | | |
|---|---|---|---|---|---|---|---|---|---|
| | Van. | AWQ | GPTQ | Van. | AWQ | GPTQ | Van. | AWQ | GPTQ |
| MATH | 55.88 | 53.96 | 53.58 | 63.52 | 62.28 | 63.12 | 66.92 | 65.72 | 65.94 |

*Note: Van. denotes the vanilla full-precision baseline.*

These results yield several key insights:

- **Larger models show greater quantization robustness.** Accuracy degradation from full-precision to quantized versions diminishes significantly as model size increases, consistent with observations from other studies. We believe because larger models possess a richer parameter space, which grants stronger robustness to quantization-induced numerical errors when mapping from high-precision to low-precision data formats.
- **Mild regularization effects appear in quantized models.** You can find an interesting result that Qwen3-14B under GPTQ slightly outperforms its vanilla counterpart, suggesting that moderate compression may enhance generalization, a phenomenon also noted in recent quantization studies. Notably, this effect has also been frequently cited as one of the underlying reasons why Quantization-Aware Training (QAT) can sometimes improve the generalization ability of post-quantized models.
- **Scalability of the "Silver Bullet" principle.** The consistently small degradation across larger models supports our hypothesis that targeted recovery using compact, well-curated data is even more effective for high-capacity models with stronger learning abilities.

## E.2   ADDITIONAL EVALUATION ON GENERAL-REASONING BENCHMARKS

To complement our analysis on mathematical reasoning, we additionally evaluate Qwen2.5-0.5B/1.5B/3B/7B-Instruct models on several widely used benchmarks that probe general science question answering, commonsense reasoning, and instruction following:

- **ARC Easy** and **ARC Challenge**: a science exam multiple-choice benchmark (grades 3–9) with two difficulty splits. Most questions provide four options and the Challenge split requires more complex reasoning (Clark et al., 2018).
- **HellaSwag**: a commonsense inference benchmark with 70k multiple-choice questions. Each item provides a scenario and four possible continuations; the distractors are adversarially generated to fool models while remaining trivial for humans (Zellers et al., 2019).
- **IFEval**: an instruction-following benchmark built from verifiable constraints (for example "write more than 400 words") that focuses on controllable instructions and reduces the bias of LLM-based judges (Zhou et al., 2023).
- **CommonSenseQA**: a 12k-question multiple-choice benchmark that requires various forms of commonsense knowledge, with one correct answer and four distractors per question (Talmor et al., 2019).

For each benchmark we compare the vanilla full-precision model with its AWQ and GPTQ quantized counterparts, and summarize the average degradation in Table 4. Compared with the larger drops on mathematical reasoning tasks, the performance drops on general commonsense and language-understanding benchmarks are substantially smaller, both in absolute score reduction and in relative

Table 4: Average accuracy degradation of Qwen2.5-Instruct models on additional benchmarks under post-training quantization. Negative values indicate performance drop compared with the corresponding full-precision models.

| Benchmark | Type | Avg. accuracy drop ↓ (points) | Avg. relative drop ↓ (%) |
|---|---|---|---|
| ARC-c | General science QA | -3.56 | -4.10 |
| ARC-e | General science QA | -3.59 | -4.76 |
| CommonSenseQA | Commonsense QA | -3.38 | -5.06 |
| HellaSwag | Commonsense | -2.35 | -3.95 |
| IFEval | Instruction following | -2.61 | -5.01 |
| **GSM8K** | **Grade-school math** | **-6.78** | **-13.21** |
| **MATH** | **Competition math** | **-10.56** | **-29.84** |

Table 5: Accuracy (%) of Qwen2.5-Instruct models on general benchmarks before and after quantization.

| Model Scale | Method | ARC-c | ARC-e | CommonSenseQA | HellaSwag | IFEval |
|---|---|---|---|---|---|---|
| 0.5B | Vanilla | 46.44 | 65.43 | 59.38 | 39.43 | 36.69 |
| | AWQ | 51.86 | 64.90 | 53.81 | 37.81 | 36.81 |
| | GPTQ | 43.39 | 50.97 | 50.37 | 36.47 | 32.01 |
| 1.5B | Vanilla | 77.97 | 89.95 | 76.00 | 62.19 | 50.96 |
| | AWQ | 72.54 | 84.13 | 73.38 | 58.62 | 47.60 |
| | GPTQ | 71.86 | 86.42 | 71.42 | 60.19 | 46.04 |
| 3B | Vanilla | 84.75 | 91.53 | 78.62 | 76.62 | 68.23 |
| | AWQ | 75.59 | 88.36 | 76.41 | 73.28 | 65.47 |
| | GPTQ | 78.64 | 89.77 | 77.56 | 73.52 | 64.87 |
| 7B | Vanilla | 86.10 | 92.59 | 84.19 | 85.18 | 77.82 |
| | AWQ | 86.44 | 93.47 | 82.47 | 84.38 | 76.86 |
| | GPTQ | 84.75 | 92.24 | 83.95 | 83.75 | 76.86 |

*Note: Inst. is an abbreviation of Instruct.*

percentage. This supports our claim that post-training quantization disproportionately affects mathematical reasoning ability while having only mild impact on general language capabilities.

The detailed per-model results are listed in Table 5. We report accuracy for each Qwen2.5-Instruct checkpoint and for each quantization method.

### E.3 CASE STATISTICS

Table 6 shows detailed statistics of all error types. The total number of cases varies slightly across models due to differences in error rates and scores.

### E.4 SUBJECT-WISE AND DIFFICULTY-WISE DEGRADATION ON MATH

To better understand how quantization-induced degradation relates to problem structure, we leverage the rich annotations in the MATH dataset, which covers multiple subject domains (such as algebra, geometry, number theory and combinatorics) and five difficulty levels (Level 1 to Level 5). For each model scale and quantization method, we compute the distribution of errors across mathematical subfields and difficulty levels. This analysis connects quantization sensitivity with different types of reasoning and provides practical guidance for deploying quantized models under varying reasoning complexities.

From the subject-wise analysis in Table 7, we observe a clear and consistent trend across all model scales: quantization-induced degradation is not uniformly distributed over mathematical domains. Subfields that involve multi-step symbolic manipulation, such as Intermediate Algebra, Precalculus and more advanced algebraic transformations, show noticeably larger performance drops for both AWQ and GPTQ. In contrast, domains that rely more on direct recall or simpler numerical reasoning remain comparatively stable. This pattern suggests that low-bit perturbations disproportionately affect

Table 6: Detailed statistics of all error types. The total number of cases varies slightly across models due to differences in error rates and scores. For full-precision models, all incorrectly answered problems are included; for quantized models, only those problems solved correctly by the full-precision model but failed after quantization are counted.

| | Method | Conceptual Errors | Method Errors | Reasoning Errors | Execution Errors | No Error | TTL |
|---|---|---|---|---|---|---|---|
| | Van. | 1622 | 313 | 427 | 380 | 28 | **2770** |
| Llama-3.1-8B-Inst. | AWQ | 286 | 86 | 5 | 136 | 4 | **517** |
| | GPTQ | 310 | 97 | 5 | 128 | 0 | **540** |
| | SQ | 199 | 58 | 7 | 102 | 3 | **369** |
| | Van. | 1760 | 369 | 387 | 387 | 82 | **2985** |
| Llama-3.2-3B-Inst. | AWQ | 317 | 91 | 1 | 107 | 1 | **517** |
| | GPTQ | 326 | 102 | 5 | 123 | 2 | **558** |
| | SQ | 236 | 65 | 4 | 88 | 3 | **396** |
| | Van. | 2521 | 515 | 491 | 491 | 40 | **4058** |
| Llama-3.2-1B-Inst. | AWQ | 287 | 87 | 6 | 108 | 0 | **488** |
| | GPTQ | 315 | 104 | 2 | 85 | 1 | **507** |
| | SQ | 196 | 85 | 4 | 70 | 0 | **355** |
| | Van. | 872 | 324 | 290 | 303 | 44 | **1833** |
| Qwen2.5-7B-Inst. | AWQ | 262 | 72 | 13 | 103 | 1 | **451** |
| | GPTQ | 267 | 82 | 11 | 116 | 4 | **480** |
| | SQ | 183 | 53 | 5 | 42 | 9 | **292** |
| | Van. | 1217 | 322 | 299 | 362 | 40 | **2240** |
| Qwen2.5-3B-Inst. | AWQ | 386 | 93 | 7 | 139 | 2 | **627** |
| | GPTQ | 351 | 120 | 7 | 130 | 3 | **611** |
| | SQ | 225 | 65 | 11 | 84 | 2 | **387** |
| | Van. | 1937 | 273 | 445 | 373 | 49 | **3077** |
| Qwen2.5-1.5B-Inst. | AWQ | 344 | 76 | 8 | 93 | 0 | **521** |
| | GPTQ | 344 | 82 | 2 | 106 | 1 | **535** |
| | SQ | 185 | 53 | 2 | 56 | 0 | **296** |
| | Van. | 2834 | 406 | 264 | 312 | 104 | **3920** |
| Qwen2.5-0.5B-Inst. | AWQ | 429 | 89 | 4 | 96 | 1 | **619** |
| | GPTQ | 521 | 59 | 3 | 70 | 1 | **654** |
| | SQ | 183 | 53 | 5 | 42 | 9 | **292** |

*Notes: Van. denotes the vanilla full-precision baseline; Inst. is an abbreviation of Instruct.*

Table 7: Distribution of errors across mathematical domains on MATH under different quantization settings. All values are percentages. All experiments are conducted on Qwen2.5-Instruct models.

| Model | Method | Number Theory | Counting & Prob. | Interm. Algebra | Algebra | Geometry | Prealgebra | Precalculus |
|---|---|---|---|---|---|---|---|---|
| | Van. | 11.59 | 10.33 | 20.48 | 18.65 | 10.30 | 16.02 | 12.63 |
| 0.5B | AWQ | 11.26 | 9.82 | 18.77 | 22.24 | 9.84 | 16.68 | 11.39 |
| | GPTQ | 11.08 | 9.80 | 18.74 | 22.57 | 9.65 | 16.71 | 11.45 |
| | Van. | 11.31 | 10.67 | 22.98 | 15.79 | 10.67 | 13.73 | 14.86 |
| 1.5B | AWQ | 11.39 | 10.26 | 20.16 | 19.16 | 10.62 | 15.72 | 12.68 |
| | GPTQ | 11.57 | 10.28 | 21.74 | 18.99 | 10.28 | 13.94 | 13.56 |
| | Van. | 10.40 | 10.35 | 24.85 | 12.91 | 11.94 | 12.07 | 17.49 |
| 3B | AWQ | 10.74 | 10.78 | 25.05 | 13.92 | 11.13 | 12.29 | 16.09 |
| | GPTQ | 11.23 | 9.98 | 24.49 | 14.58 | 11.20 | 12.30 | 16.22 |
| | Van. | 7.96 | 10.31 | 26.28 | 11.75 | 12.87 | 11.91 | 18.91 |
| 7B | AWQ | 9.58 | 9.88 | 25.89 | 11.87 | 12.22 | 11.67 | 18.90 |
| | GPTQ | 8.56 | 10.50 | 27.13 | 11.20 | 12.39 | 11.60 | 18.62 |

*Note: Van. denotes the vanilla full-precision baseline.*

tasks that require long-range dependency tracking and precise arithmetic transformations, which is consistent with the step-level error analysis in the main paper.

From the difficulty-level analysis in Table 8, we see a monotonic increase in degradation as problem difficulty grows. Level 1 and Level 2 questions exhibit only marginal changes after quantization, whereas degradation becomes much more pronounced for Levels 3 to 5. For Level 5 in particular, the gap between full-precision and quantized models can exceed 6–10 percentage points even for larger models.

These findings provide empirical evidence for the following points:

Table 8: Distribution of errors across difficulty levels on MATH under different quantization settings. All values are percentages.

| Model | Level 1 | Level 2 | Level 3 | Level 4 | Level 5 |
|---|---|---|---|---|---|
| Qwen2.5-0.5B-Inst. Van. | 5.72 | 15.41 | 21.39 | 26.04 | 31.44 |
| Qwen2.5-0.5B-Inst. AWQ | 7.53 | 17.29 | 22.43 | 24.99 | 27.75 |
| Qwen2.5-0.5B-Inst. GPTQ | 7.17 | 17.35 | 22.44 | 24.99 | 28.05 |
| Qwen2.5-1.5B-Inst. Van. | 4.38 | 10.42 | 20.30 | 25.59 | 35.71 |
| Qwen2.5-1.5B-Inst. AWQ | 4.89 | 16.57 | 21.37 | 25.31 | 31.48 |
| Qwen2.5-1.5B-Inst. GPTQ | 4.41 | 14.22 | 21.03 | 23.82 | 36.53 |
| Qwen2.5-3B-Inst. Van. | 3.39 | 12.20 | 19.03 | 25.73 | 39.65 |
| Qwen2.5-3B-Inst. AWQ | 3.30 | 12.21 | 20.12 | 25.86 | 38.50 |
| Qwen2.5-3B-Inst. GPTQ | 3.16 | 15.27 | 19.52 | 26.50 | 35.55 |
| Qwen2.5-7B-Inst. Van. | 3.42 | 12.45 | 19.07 | 26.01 | 39.05 |
| Qwen2.5-7B-Inst. AWQ | 3.34 | 12.02 | 18.75 | 25.79 | 40.10 |
| Qwen2.5-7B-Inst. GPTQ | 3.38 | 12.00 | 19.31 | 25.19 | 40.12 |

*Notes: Van. denotes the vanilla full-precision baseline; Inst. is an abbreviation of Instruct.*

- Conceptually demanding and algebraically heavy subfields of MATH are especially vulnerable to precision reduction.
- Higher-difficulty problems, which require deeper chains of reasoning, tend to accumulate quantization noise and errors more severely.

Overall, this analysis further supports our claim that post-training quantization disproportionately affects mathematical reasoning compared with general language understanding. We include these tables and observations in the appendix for completeness.

## F    CAPABILITY RESTORATION RESULTS

Table 9: Capability restoration results on GSM8K, MATH500, and MMLU benchmarks across different model scales using our curated *Silver Bullet* datasets. **Full Precision** refers to the full-precision model after format alignment. **BF** indicates performance before restoration, while **AF** shows performance after applying our restoration pipeline.

| Quantization | Task | Llama-3-Inst. | | | Qwen2.5-Inst. | | | |
|---|---|---|---|---|---|---|---|---|
| | | 1B | 3B | 8B | 0.5B | 1.5B | 3B | 7B |
| Full Precision | GSM8K | 38.44 | 71.34 | 76.88 | 42.99 | 61.87 | 76.04 | 75.51 |
| | MATH500 | 18 | 32.4 | 36.4 | 16.6 | 22.2 | 39 | 41.6 |
| | MMLU | 45.14 | 61.81 | 68.62 | 45.49 | 59.71 | 65.1 | 73.32 |
| | AVG | **33.86** | **55.18** | **60.63** | **35.03** | **47.93** | **60.05** | **63.48** |
| AWQ-BF | GSM8K | 35.03 | 70.58 | 77.1 | 27.9 | 53.15 | 70.36 | 77.63 |
| | MATH500 | 13.8 | 29.6 | 33.2 | 8.2 | 21 | 29 | 42.4 |
| | MMLU | 43.26 | 60.08 | 67 | 42.65 | 57.65 | 63.16 | 71.77 |
| | AVG | **30.70** | **53.42** | **59.10** | **26.25** | **43.93** | **54.17** | **63.93** |
| GPTQ-BF | GSM8K | 32.15 | 69.67 | 76.27 | 25.02 | 57.09 | 68.54 | 81.12 |
| | MATH500 | 15.4 | 26.4 | 33.6 | 8.6 | 21.8 | 31.2 | 41.6 |
| | MMLU | 42.07 | 59.49 | 66.44 | 42.91 | 57.86 | 62.09 | 71.49 |
| | AVG | **29.87** | **51.85** | **54.94** | **25.51** | **45.58** | **53.94** | **64.74** |
| AWQ-AF | GSM8K | 40.49 | 74.3 | 80.14 | 26.38 | 56.86 | 68.84 | 76.42 |
| | MATH500 | 15.2 | 36.8 | 34.6 | 9.4 | 26.4 | 38.4 | 45.6 |
| | MMLU | 43.72 | 60.57 | 67.67 | 43.99 | 59.43 | 64.8 | 73.1 |
| | AVG | **33.14** | **57.22** | **60.80** | **26.59** | **47.56** | **57.35** | **65.04** |
| GPTQ-AF | GSM8K | 37.83 | 73.01 | 79.68 | 25.93 | 55.65 | 75.21 | 76.88 |
| | MATH500 | 18.2 | 33 | 36 | 8.4 | 25.2 | 40.6 | 46 |
| | MMLU | 42.29 | 59.9 | 67.23 | 44.15 | 59.43 | 63.63 | 72.56 |
| | AVG | **32.77** | **55.30** | **60.97** | **26.16** | **46.76** | **59.81** | **65.15** |

As shown in Table 9, after capability restoration using our *Silver Bullet* dataset, the quantized 4-bit models not only recover but even surpass the performance of their full-precision counterparts on the MATH benchmark. Meanwhile, performance on GSM8K remains stable, and accuracy on MMLU—a diverse benchmark covering various complex reasoning tasks—is also preserved. These results demonstrate that our *Silver Bullet* dataset effectively restores mathematical reasoning capabilities without compromising general-purpose abilities, highlighting both the effectiveness and generalizability of our approach.

## G    THE USAGE OF LLM

In this work, Large Language Models (LLMs) were used as auxiliary tools to support our research process, but not to generate novel scientific content. Specifically, their usage includes:

- **Editing and polishing.** LLMs were employed for minor grammar checking, improving clarity, and rephrasing sentences for readability in the manuscript. All scientific ideas, methodology, and experiments were designed and written by the authors.
- **Facilitating annotation.** During the construction of our automated error-assessment pipeline, LLMs were used as expert judges to classify error types in reasoning traces. Their outputs were combined via majority voting and, when necessary, verified by human annotators to ensure reliability.
- **Experiment assistance.** LLMs were queried to simulate baseline reasoning traces for building our contrastive "Silver Bullet" datasets, which were later curated, filtered, and validated by the authors. This step complements human effort by accelerating the generation of positive examples.

We emphasize that all key contributions—including research ideas, methodology design, experimental execution, and analysis—were conceived and implemented by the authors.

