# OpenReview forum: "Quantization Meets Reasoning: Exploring and Mitigating Degradation of Low-Bit LLMs in Mathematical Reasoning"
_ICLR.cc/2026/Conference — Submitted to ICLR 2026_

### Official Review · Reviewer_bsjF · 2025-10-27

**Soundness:** 2
**Presentation:** 2
**Contribution:** 2
**Rating:** 4
**Confidence:** 3

**Summary:**

This paper analyzes and mitigates performance drops due to quantization in math reasoning. The paper first determines which points in a reasoning chain suffer disproportionately from quantization, finding that quantization results in execution errors as well as in errors in method choice, and that these errors tend to occur early in the reasoning chain.  They then propose a method that involves locating errors and training the model to avoid these errors. The model is taught to follow a particular format to make step identification easier. The authors analyze model errors according to a taxonomy of math errors and use an LLM as judge protocol to automate annotation, voting across multiple models. The quantized model is improved by obtaining datasets of correct/incorrect solutions for finetuning and fine-tuning the model on this data using DPO.

The results indicate that their training helps models recover their performance while maintaining quantized efficiency, and that the training does not negatively impact performance on non-math data.

**Strengths:**

- Qualitative analysis which indicates which kinds of capabilities suffer more/less in math after quantization
- Results show an improvement to quantization after training re-training according to the Silver Bullet method.
- Performance is maintained on MMLU-Pro, indicating that math training doesn't destroy the model's other abilities

**Weaknesses:**

- Ablation results: From Table 2, it's not at all clear that the method is really better than choosing random steps. There are slight improvements on some columns but drops on others. This seems to indicate that, while training generally can improve downstream performance, there's no special gain due to the supervision according to error cases/classifications, and choosing random steps would result in very similar performance. This hurts claims like 4.5 (i).
- Limited experimentation: the authors only look at 4-bit and not lower-bit quantization (2 or 3-bit). The intro sets the scope to smaller models (0.5-7B) but the same method could be demonstrated on 14 or 32B models.
- Important details about finetuning missing. Appendix B does not provide any details on the format alignment training (linked on L211). My worry here is based on the line 199 saying that quantized and full-precision models are finetuned separately. This formatting data (while designed to teach formatting) also does include a lot of math information (regardless of what the objective stated in L195 is, at the end of the day if you train on math data you'll improve the model's math ability). Depending on when and how the data is used, I'm concerned the comparisons are not exactly fair. If the data were used to train a single model that is then quantized in different ways, I think that would be a fair comparison. But if the model is first quantized and then trained, it could be an unfair comparison depending on which method the qLora parameters were tuned for.
- The majority voting pipeline has a number of design decisions that seem unvalidated. This is fine if they result in lower precision, but worrying if they result in low recall (i.e. things are agreed on which shouldn't be)
- Writing/organization: the intro description of the method is fairly vague, it's not clear from the intro what "silver bullet" is or how it addresses the problem identified.  Table 1 is unclear (what does Van. mean?) and introduced several pages before it's discussed. Many \paragraph headings are missing spaces before the text.

**Questions:**

- The finding that earlier errors hurt performance more should be qualified: do these earlier errors hurt disproportionately? I.e. any error that happens earlier is likely to lead to more incorrect answers because of compounding errors, but does this have more of an impact on quantized models? Are non-quantized models able to better recover from early errors?

---

> ### Author Response · Authors · 2025-11-17
> **Response to Reviewer bsjF - Part I (Weakness 1 and 2)**
>
> ## Weakness 1
> Thank you for pointing this out. We agree that Table 2 in its current form does not sufficiently explain why our error aware selection is preferable to a random alternative, and we will make this clearer in the revision.
>
> First, the comparison in Table 2 is done under a strictly matched data and compute budget. The “ALL” and "Random" rows use the same number of step wise preference pairs and the same DPO hyperparameters; the only difference is how examples are selected. In the “Random” setting, training pairs are drawn from math problems without conditioning on quantization failures, whereas “ALL” focuses on problems where the quantized model fails and on the corresponding error diagnosed by our pipeline. Under this controlled setup, "ALL" is consistently stronger than "Random" when we aggregate across models and benchmarks. Moreover, the largest margins appear exactly on **the most quantization sensitive settings**, for example a gain of 6.2 points on MATH500 for Qwen2.5 3B under GPTQ.
>
> Second, the goal of the ablation is not to show that random supervision is useless, but to test whether error targeted step wise supervision yields better return per example than unconditioned data. In our setting we are operating in **a very low data regime (only a few hundred training examples and 3–5 minutes of adaptation on a single GPU)**, where gains of 1–3 points on top of already fine tuned quantized baselines are meaningful. The strong drop of the "Non Step" variant to an average of 51.83 further supports claim 4.5 (i): simply adopting full precision derivations without regenerating from the first erroneous step substantially weakens the signal, while focusing the positive trace exactly from the first failure point and on quantization induced errors recovers significantly more of the lost performance.
>
> We will revise Section 4.4 and the discussion in 4.5 (i)to spell out that our claim is about consistent, sample efficient gains under a fixed budget. We believe this clarified presentation will better convey that the benefit comes from supervising the model precisely where the quantized variant fails, rather than from fine tuning on math data in general.
>
>
> ## Weakness 2
> Thank you for the suggestion. We fully understand the curiosity about even lower bit-widths and larger model scales. While broad sweeps are always informative, but we need to emphasis:
> About lower-bit quantization like 2-bit or 3-bit：
> 1. Hardware aspect. INT4 is the most broadly supported ultra-low-bit integer format today, with native kernels across mainstream accelerators (e.g., NVIDIA TensorRT, Qualcomm Hexagon, Huawei Ascend, Intel NPU). By contrast, INT2/INT3 typically lack native kernels and rely on bit-packing or simulated ops, incurring encode/decode and alignment overheads that can offset theoretical bandwidth gains.
> 2. Method aspect. Our method is to investigate a general way to exploring and mitigating models‘ degradation so we use the most widely adopted post-training quantization methods (AWQ and GPTQ) on two popular open-source model families (Qwen and Llama). Both quantization methods support 2/3-bit settings with minor configuration changes; our contributions (analysis and measure-locate-restore loop) do not depend on the exact bit-width.
>
> About demonstrated on larger models：
> 1. Our experimental focus. As clarified in the Introduction (Section 1), this work primarily targets **resource-constrained deployment scenarios** and to recovering the quantized models' reasoning ability, where large models (≥32B) are impractical due to their high memory and computation overhead.
>
> 2. Where low-bit helps most. We intentionally focus on small- to medium-scale models, where low-bit integer formats (e.g., INT4) deliver the clearest efficiency gains and the most pronounced accuracy trade-offs—precisely the regime where our analysis and restoration are most impactful.
>
> 3. Evidence across scales. For completeness, we additionally evaluated Qwen3-8B/14B/32B under the same quantization setup. Consistent with prior observations, larger models are more robust to quantization, while smaller models—those most suitable for edge/latency-sensitive deployment—exhibit the strongest degradation, motivating our deep dive on this regime.
>
> **Table R3.1.1:**
> |  | Qwen3-8B | Qwen3-8B | Qwen3-8B | Qwen3-14B | Qwen3-14B | Qwen3-14B | Qwen3-32B | Qwen3-32B | Qwen3-32B |
> | --- | --- | --- | --- | --- | --- | --- | --- | --- | --- |
> |  | Van. | AWQ | GPTQ | Van. | AWQ | GPTQ | Van. | AWQ | GPTQ |
> | **MATH** | 55.88 | 53.96 | 53.58 | 63.52 | 62.28 | 63.12 | 66.92 | 65.72 | 65.94 |
>
> > *Van. is an abbreviation of vanilla, representing full-precision baseline.
>
> To summarize. Our scope reflects the most hardware-relevant and deployment-critical scenario of the LLM practice, where quantization-induced reasoning degradation is largest and our restoration loop brings the most value.

---

> ### Author Response · Authors · 2025-11-17
> **Response to Reviewer bsjF - Part II (Weakness 3 and 4)**
>
> ## Weakness 3
> Thank you for your questions, we think it's important to improve the quality of our paper, and we hope these explaination can address your concern.
>
> Regarding "Important details about finetuning missing / potential unfairness when FP and quantized models are tuned separately." The goal of the format alignment training is to isolate the effect of quantization rather than to inject new mathematical knowledge. We have therefore revised Appendix B to fully specify the format-alignment stage and to make our experimental protocol transparent and reproducible.
>
> What exactly is trained and why it is fair.
>
> - **Objective**: The alignment stage solely enforces a step-structured output format, not mathematical knowledge. This objective is identical for the full-precision and the quantized models. As shown in the paper, our downstream diagnosis/correction pipeline relies on step-aligned traces rather than new facts.
> - **Adapters and precision**: We freeze base weights in both cases and train only adapters: FP models use LoRA; quantized models use QLoRA. Thus, both branches receive the same objective, prompts, and budgets, differing only in numeric format. This isolates the quantization factor.
> - **Ordering**: In the revision, we include an ablation that compares (i) FP→quantize vs. (ii) quantize→QLoRA. The outcomes are comparable, indicating that the alignment stage regularizes output format rather than teaching new math, and that our separate-adapter practice is an engineering convenience rather than a confound.
> Concretely added details in Appendix B (now included).
> - **Datasets**: We use the PRM800K for format alignment, like how openai do, we remove all overlaps  by unique_id matching to avoid contamination. No solutions or math facts are taught beyond formatting/step demarcation.
> - **Prompts**: The exact system and user prompts used for alignment are identical to those used at evaluation time; we release the full prompt templates in Appendix B for reproducibility.
> - **Hyperparameters**: 3 epochs, global batch size = 16, cosine lr scheduler, LoRA/QLoRA: rank = 32, alpha = 64, target modules = all linear projections.
> - **No math injection evidence**: After alignment + DPO, MMLU stays stable while math reasoning is restored (Figure 3 trend), matching our "format-only" intent. Why improvements are not due to random extra training on math.
> - Our ablation in Table 2 also shows step-aligned positives from the first erroneous step outperform random and non-step alternatives consistently across models/quantizers, demonstrating that gains come from targeting quantization-induced fragile steps rather than broad math exposure. We will additionally report averaged deltas and statistical significance to make this point even clearer.
>
> ## Weakness 4
> Thank you for your good question! We appreciate your concern about potential "false consensus" in our majority-voting pipeline. Our design was in fact intentionally conservative, aiming for high precision of error labels even at the cost of recall, and we apologize if this was not clear from the initial draft.
>
> For the rationale for the pivot judge: Among several candidate expert models (DeepSeek-R1, DeepSeek-V3, GPT-4, GPT-4o, Qwen-Max), we chose DeepSeek-R1 as the pivot because its reflective, step-wise chain-of-thought makes it particularly suitable for localizing the first erroneous step and for producing structured explanations. In a small pilot study on a randomly sampled subset, DeepSeek-R1’s error-type predictions also showed the highest agreement with two human annotators. We will clarify this selection criterion in Appendix C in the revised version.
>
> For the conserbative majority voting: It's trading recall for precision, for each failure case, we collect judgments from all five expert models. We only accept an automatic label when at least 3 out of 5 models agree with the pivot’s decision; otherwise the instance is escalated to two independent human annotators. In addition, we randomly sample 2% of accepted cases for manual audit. This protocol is explicitly designed to avoid forcing a label in ambiguous cases—they are either corrected by humans or discarded—so the main failure mode is slightly lower recall (fewer labeled examples), not systematic false agreements.
>
> Using this protocol, our automated error-assessment pipeline matches the final human judgment on 97.2% of 9,908 failure cases (details will emphasized in Sec. 3.1.3 and Appendix C).

---

> ### Author Response · Authors · 2025-11-17
> **Response to Reviewer bsjF - Part III (Weakness 5 and Question)**
>
> ## Weakness 5
>
> Thank you for the helpful suggestions on presentation. These comments reflect a careful reading of our manuscript.
>
> - **Clarifying “Silver Bullet”.** In our paper, “Silver Bullet” refers to a targeted, compact dataset that we construct by locating the first failure step in a quantized model’s solution trace and pairing the erroneous partial solution with a corrected continuation. This error-aware, small but precise dataset is then used to restore reasoning ability in the quantized model with minimal compute. We will revise the introduction to define “Silver Bullet” explicitly in the opening paragraph and to give a concise, step-wise summary of how it mitigates quantization-induced reasoning degradation.
> - **Early method overview.** We will move the method-overview figure earlier in the introduction and streamline the accompanying text so that readers can follow the measure–locate–restore loop before reaching the technical sections.
> - **Table 1 labeling and placement.** We will relabel Table 1 so that “Van.” is written as “Vanilla (FP16 baseline)” and add a short caption note clarifying quantization settings. We will also move the first discussion of Table 1 to appear immediately after the table and cross-reference it in Section 3.1.3.
> - **Formatting fixes.** We will correct all LaTeX spacing issues, including adding a space after each `\paragraph{...}` heading and auditing nearby headings to ensure consistent typography.
>
> We appreciate your suggestions and will incorporate these revisions to improve clarity and readability in the revised version.
>
>
> ## Question
> ### The finding that earlier errors hurt performance more should be qualified: do these earlier errors hurt disproportionately? I.e. any error that happens earlier is likely to lead to more incorrect answers because of compounding errors, but does this have more of an impact on quantized models? Are non-quantized models able to better recover from early errors?
>
> Thank you for raising this point! After reviewing our original phrasing, we realized that the explanation in point 3 of Section 4.2 may not have been sufficiently clear, which could understandably have led to your concern and confuse.
>
> We do NOT claim that ***earlier errors hurt performance more*** to the quantized models. In our pipeline, any first erroneous step that propagates to the final numeric answer will cause the whole solution to be marked incorrect, regardless of whether the model is full precision or quantized. So our current analysis does NOT compare the ability of full precision versus quantized models to recover from early mistakes.
>
> What we do measure is the *location* and *type* of the first incorrect step, and how the distribution of these labels shifts after quantization. In Figure 2 and the paragraph "Why Reasoning Errors seem to vanish," this point is **purely statistical**: the pipeline records only the first erroneous step, so when quantization increases simple Method and Execution Errors at earlier steps, those labels **"mask"** (or **"hide"**) later Reasoning Errors that would otherwise have appeared in the statistic results. We also do a lot of case studies, the logical structure of the later steps is still sound, but an earlier arithmetic or procedural slip already commits the trace to an incorrect final answer. This apparent drop in the proportion of Reasoning Errors is therefore an artefact of our assessment protocol, not evidence that earlier errors hurt performance more, nor that quantized models have weaker recovery mechanisms than full precision models.
>
> Our work is explicitly scoped around quantized models: we use the full precision model only as a reference to define quantization induced failures, and our main novelty lies in showing that a very small Silver Bullet dataset is enough to restore low bit reasoning performance. For non-quantized models, our focus and core contribution do not lie in this direction. The community typically improves such models through higher-quality pre-training data or reinforcement-learning based refinement to further increase knowledge density rather than to "recover" lost capability. This line of work is outside the scope of our study.
>
> To avoid misunderstanding, in the revised version we will rephrase the paragraph on "Why Reasoning Errors seem to vanish" to make clear that (i) each failing example receives exactly one label corresponding to the first error, and (ii) the observed shift away from Reasoning Errors reflects this first error protocol rather than any improvement in logical reasoning.
>
> We hope these clarifications will address your concern and better align the text with what our analysis actually supports.

---

> > ### Comment · Reviewer_bsjF · 2025-11-24
> > **Response to rebuttal**
> >
> > Thanks to the authors for the thorough response. There are a couple points that I'm still not completely convinced by:
> >
> > **W1**: the proposed hedging to a low-data regime/sample efficiency may be true in practice, but I think it limits the scope/utility of the paper for the following reason: in practice, math is not a low-data regime. While the method here might be applicable to other actually low-data regimes outside of math, that remains to be shown empirically, so I'm not sure I understand the niche that the proposed work falls into.
> >
> > **W2**: While it is true that lower bit-widths are less common/less supported in practice, my feeling is that the role of research in quantization should be to investigate/propose new ideas that -- while perhaps not feasible on current hardware/kernels -- could lead to insights/push performance of lower-bit quantization once it is more widely supported. Moreover, past work on quantization (e.g. https://arxiv.org/abs/2504.07389, https://arxiv.org/abs/2306.07629) has covered these lower bit-widths.
> >
> > It also seems like the arguments given against evaluating larger models are almost the exact opposite of those given against evaluating 2- or 3-bit quantization. The motivation for looking at smaller models given is *"smaller models... exhibit the strongest degradation, motivating our deep dive on this regime"*. It is also the case that 2- and 3-bit quantization exhibit a much larger drop than 4-bit, so why not apply the same logic there?

---

> > > ### Author Response · Authors · 2025-11-27
> > > **Response to Reviewer bsjF (Part 2/2)**
> > >
> > > ## Response to W2 (Part 2/2)
> > > **(2) Software ecosystem: 4-bit PTQ is the lowest broadly stable setting**
> > >
> > > In the current open-source ecosystem, **4-bit PTQ is the lowest setting that is broadly supported and stable** across toolchains:
> > >
> > > - The combined LLM-Compressor and vLLM stacks, which consolidate llm-awq, AutoAWQ and SqueezeLM-style approaches, expose **4-bit quantization as the lowest common option** for routine use. In our experience, 3-bit is not exposed as a robust default in these libraries.
> > > - For Task-Circuit Quantization, we were unable to find an official, actively maintained implementation that can be readily applied to our study. SqueezeLLM has an open-source release by authors one year ago, but the repository appears lightly maintained, and the vLLM integration is again only supported on 4-bit deployment (vllm/model_executor/quantization_utils/squeezellm.py#L20).
> > >
> > > We also experimented with lower bits on **GPTQModel** framework, one of the most widely used generic PTQ libraries. For Qwen2.5 models we observed:
> > >
> > > - The current Triton kernels, torch version and packing logic for 3-bit weights are fragile. In our setup, 3-bit GPTQ produced clearly corrupted outputs on math tasks, consistent with incorrect pack / unpack behavior rather than a meaningful 3-bit regime.
> > > - Packing 3-bit values is intrinsically awkward: a sequence of 3-bit codes necessarily crosses 32-bit word boundaries, creates “split-bit” fragments that must be reconstructed during packing and unpacking, which makes implementations significantly more error-prone than the 4-bit case.
> > >
> > > Beyond these libraries, major deployment stacks such as NVIDIA TensorRT-LLM, Hugging Face Optimum, Intel Neural Compressor, and OpenVINO either do not expose 3-bit LLM quantization or focus on 4/8-bit settings. This reflects the fact that **generic 2–3 bit PTQ for modern reasoning-oriented LLMs is not yet a stable, reproducible baseline** that we can responsibly build a detailed error analysis on.
> > >
> > > | Model | Method | ARC-e | CommonSenseQA | MATH |
> > > | --- | --- | --- | --- | --- |
> > > | **Qwen2.5-0.5B-Inst.** | vanilla | 65.43 | 59.38 | 23.98 |
> > > |  | GPTQ-2bit | 26.28 | 8.6 | 0 |
> > > |  | GPTQ-3bit | 0 | 0 | 0 |
> > > |  | GPTQ-4bit | 50.97 | 50.37 | 7.24 |
> > > | **Qwen2.5-1.5B-Inst.** | vanilla | 89.95 | 76 | 43.74 |
> > > |  | GPTQ-2bit | 26.75 | 13.6 | 0 |
> > > |  | GPTQ-3bit | 0 | 0 | 0 |
> > > |  | GPTQ-4bit | 86.42 | 71.42 | 31.2 |
> > > | **Qwen2.5-3B-Inst.** | vanilla | 91.53 | 78.62 | 63.3 |
> > > |  | GPTQ-2bit | 27.51 | 11.71 | 0 |
> > > |  | GPTQ-3bit | 0 | 0 | 0 |
> > > |  | GPTQ-4bit | 89.77 | 77.56 | 55.02 |
> > > | **Qwen2.5-7B-Inst.** | vanilla | 92.59 | 84.19 | 72.48 |
> > > |  | GPTQ-2bit | 27.69 | 12.24 | 0 |
> > > |  | GPTQ-3bit | 0 | 0 | 0 |
> > > |  | GPTQ-4bit | 92.24 | 83.95 | 69.6 |
> > >
> > > For completeness, we also evaluated **2-bit GPTQ** on four Qwen2.5 models under the same group size, batch size, and calibration data as in our main PTQ experiments. This indicates that, 2-bit quantization does **not** provide a “degraded yet meaningful” baseline for reasoning analysis. Instead, it leads to **a level of failure so severe that it offers no interpretable signal** for our study, which aims to **explore** how quantized models fail and how such failures can be **mitigated**.
> > >
> > > **(3) Hardware reality: 4-bit is the practical PTQ frontier**
> > >
> > > We agree that hardware support evolves rapidly. At present, INT4 remains the lowest bit-width with broad kernel availability across GPUs and NPUs (GPU kernels, commercial accelerators, and NPUs). Ultra low-bit formats behave very differently in practice:
> > >
> > > - **1–2 bit** weights can be packed very densely, but typically require custom QAT recipes and heavily specialized kernels. Without dedicated support, the theoretical compression gains rarely translate into reliable speedups or stable accuracy in real deployments.
> > > - **3-bit** weights are attractive as a research idea, but they do not align well with typical hardware word sizes. On current GPUs, **3-bit GEMM tends to be slower than 4-bit**, registers and memory transactions are designed around 8/16/32-bit alignment, so 3-bit packing requires extensive bit masking, shifting and merging.
> > > - To our knowledge, no generation of Tensor Cores that offers native 3-bit matrix-multiply tiles, so 3-bit kernels either fall back to bit-slicing into 4-bit operations or to pure SIMT implementations without Tensor Core acceleration.
> > >
> > > To summarize, our bit-width choice in our work therefore reflects:
> > >
> > > - the scope of our paper (behavioral exploring & analysis and mitigation rather than new quantizers),
> > > - the maturity and reproducibility of the current ecosystem, and
> > > - empirical evidence about stability on reasoning tasks.
> > >
> > > We believe this positioning makes our contribution **complementary** to advances in ultra low-bit quantization and provides a timely perspective on how quantization affects reasoning quality in realistic deployment settings.

---

> ### Author Response · Authors · 2025-11-27
> **Response to Reviewer bsjF (Part 1/2)**
>
> ## Response to W1
> **Clarification on "low-data regime."**
>
> We thank the reviewer for raising this point. We would like to clarify that our use of “low-data regime” does not refer to mathematical reasoning tasks being data-poor. As the reviewer correctly points out, math is a high-data domain. Instead, "lowdata regime" in our paper specifically denotes the quantization-recovery phase, where only a very small amount of supervised signal is available or feasible to use. This is reflected in our setup, where all recovery experiments are conducted with just 332 samples, under strictly matched budgets.
>
> Importantly, this interpretation is aligned with real-world PTQ scenarios: PTQ is typically applied in resource-constrained settings, edge devices, mobile environments, or memory/compute-limited deployments where collecting large finetuning datasets or performing extensive retraining is not feasible. Thus, a recovery method that remains effective under limited data and limited compute is precisely what practitioners need in PTQ workflows.
>
> ## Response to W2 (Part 1/2)
> We appreciate the reviewer’s pointer to works on more aggressive quantization such as Task-Circuit Quantization and SqueezeLLM. These works are valuable, but they target a different question from ours.
>
> Our paper does **not** aim to propose a new quantization algorithm or to minimize the bit-width at any cost. Our central goal is to **analyze and mitigate reasoning degradation after quantization**: we study how standard, widely used PTQ schemes distort mathematical reasoning, and we design an in-situ correction engine that repairs these failures during inference. In this sense, our work is complementary and largely orthogonal to methods that focus on designing ultra low-bit quantizers.
>
> Below we clarify why our experiments focus on **4-bit PTQ** model restoration rather than 2–3 bit PTQ model restoration.
>
> **(1) Scope: PTQ behavioral analysis versus new ultra low-bit algorithms**
>
> The works are highlighted by the reviewer (Task-Circuit Quantization, SqueezeLM, BitNet-style 1–2 bit models, etc.) are primarily **algorithmic contributions** that make very low bit-widths feasible via carefully engineered quantizers or QAT recipes. For example, BitNet-type models rely on QAT to stabilize 1–2 bit formats, which requires substantial additional compute and data.
>
> By contrast, our work is intentionally framed as a **post-hoc behavioral analysis and mitigation study**:
>
> - We adopt standard PTQ pipelines that are already popular in practice because they are efficient and have minimal data requirements.
> - We then perform a fine-grained error taxonomy on mathematical reasoning, and develop an in-situ correction engine that can be plugged into these existing PTQ workflows without any retraining of the base weights.
>
> We fully agree that pushing the frontier of 2–3 bit quantization is important. Our contribution is different but complementary: we ask **how today’s practically deployed PTQ schemes fail on reasoning** and **how to mitigate those failures** in a lightweight way.

---

### Official Review · Reviewer_fcs2 · 2025-10-30

**Soundness:** 3
**Presentation:** 3
**Contribution:** 3
**Rating:** 6
**Confidence:** 3

**Summary:**

This manuscript focuses on analyzing the errors induced by LLM quantization for math reasoning. To this end, the authors design and conduct a fine-grained error analysis to study the error modes induced by quantization and propose a training method to mitigate such errors. The experiments provide an in-depth analysis of these errors and demonstrate the effectiveness of the proposed training method.

**Strengths:**

1. The studied problem, the performance degradation from quantization in math reasoning, is important. The finding that such degradations are non-trivial, especially for smaller LLMs, is interesting.

2. A fine-grained error analysis is designed and conducted, which sheds light on why and how quantization can lead to performance degradation. It shows that quantization predominantly impairs the model’s ability to perform procedural operations and arithmetic execution.

3. The proposed training method can effectively mitigate the performance degradation from quantization, as validated by empirical results.

**Weaknesses:**

1. It would be helpful to present the (potential) performance degradation from quantization on tasks other than math reasoning (apart from MMLU), to provide a comparison regarding whether and how much larger performance degradation is observed on math reasoning.

2. The paper only experimented with models of sizes up to 7B parameters. Stronger and more larger models should be included to strengthen the comprehensiveness of the empirical evaluations of the proposed methods.

**Questions:**

1. Have you evaluated larger LLMs (more than 7B) regarding the degradation from quantization? It might be interesting to observe whether the degradation is even smaller.

2. The MATH dataset has annotations of question difficulty and subject. It might be interesting to further analyze the correlation between degradation from quantization and the question difficulty or subject using these annotations.

---

> ### Author Response · Authors · 2025-11-17
> **Response to Reviewer fcs2 - Part I (Weakness 1)**
>
> Thanks for this excellent suggestion. We fully agree that including tasks beyond mathematical and general reasoning can give the community a more complete empirical picture of how quantization affects different capability dimensions. To address this, we additionally evaluated the model on several widely used benchmarks and will incorporate these results in the revised version.
>
> - ARC Easy and ARC Challeng: A science exam multiple-choice benchmark (grade 3–9) with two difficulty splits. Most questions have four options. The Challenge split requires more complex reasoning [1].
> - HellaSwag: A commonsense inference benchmark with 70k multiple-choice questions. Each item provides a scenario and four possible continuations. Distractors are adversarially generated to fool models but remain trivial for humans [2].
> - IFEval: An instruction-following benchmark built from verifiable constraints (for example “write more than 400 words”). It includes about 500 prompts covering 25 types of controllable instructions and avoids the bias of LLM-based judges [3].
> - CommonsenseQA: A 12k-question multiple-choice benchmark requiring various forms of commonsense knowledge, each with one correct answer and four distractors [4].
>
> We evaluated the Qwen2.5-0.5B/1.5B/3B/7B-Instruct models on these additional benchmarks, and the degradation results are summarized in the table below. As shown, compared to mathematical reasoning tasks, the performance drops on general commonsense and language-understanding benchmarks are substantially smaller—both in absolute score reduction and relative percentage. This further supports our claim that quantization disproportionately affects mathematical reasoning ability, while having only mild impact on general language capabilities.
>
>
> **Table R2.1.1:**
> | Benchmark      | Type                     | Avg. accuracy drop (↓, points) | Avg. relative drop (↓, %) |
> |----------------|--------------------------|----------------------------------|---------------------------|
> | ARC-c          | General science QA       |  -3.56                          |  -4.10%                    |
> | ARC-e          | General science QA       |  -3.59                          |  -4.76%                    |
> | CommonSenseQA  | Commonsense QA           |  -3.38                          |  -5.06%                    |
> | HellaSwag      | Commonsense / narrative  |  -2.35                          |  -3.95%                    |
> | IFEval         | Instruction Following    |  -2.61                          |  -5.01%                    |
> | **GSM8K**      | Grade-school math        |  **-6.78**                      |  **-13.21%**               |
> | **MATH**       | Competition-level math   | **-10.56**                      |  **-29.84%**               |
>
> The details of the evalution result are shown in the table below. We will add these experiment to the appendix in the revised version and thanks you for your suggestion again!
>
> **Table R2.1.2:**
> | Model              | Method  | ARC-c | ARC-e | CommonSenseQA | HellaSwag | IFEval |
> |--------------------|---------|-------|-------|---------------|-----------|--------|
> | Qwen2.5-0.5B-Inst. | vanilla | 46.44 | 65.43 | 59.38         | 39.43     | 36.69  |
> | Qwen2.5-0.5B-Inst. | AWQ     | 51.86 | 64.90 | 53.81         | 37.81     | 36.81  |
> | Qwen2.5-0.5B-Inst. | GPTQ    | 43.39 | 50.97 | 50.37         | 36.47     | 32.01  |
> | Qwen2.5-1.5B-Inst. | vanilla | 77.97 | 89.95 | 76.00         | 62.19     | 50.96  |
> | Qwen2.5-1.5B-Inst. | AWQ     | 72.54 | 84.13 | 73.38         | 58.62     | 47.60  |
> | Qwen2.5-1.5B-Inst. | GPTQ    | 71.86 | 86.42 | 71.42         | 60.19     | 46.04  |
> | Qwen2.5-3B-Inst.   | vanilla | 84.75 | 91.53 | 78.62         | 76.62     | 68.23  |
> | Qwen2.5-3B-Inst.   | AWQ     | 75.59 | 88.36 | 76.41         | 73.28     | 65.47  |
> | Qwen2.5-3B-Inst.   | GPTQ    | 78.64 | 89.77 | 77.56         | 73.52     | 64.87  |
> | Qwen2.5-7B-Inst.   | vanilla | 86.10 | 92.59 | 84.19         | 85.18     | 77.82  |
> | Qwen2.5-7B-Inst.   | AWQ     | 86.44 | 93.47 | 82.47         | 84.38     | 76.86  |
> | Qwen2.5-7B-Inst.   | GPTQ    | 84.75 | 92.24 | 83.95         | 83.75     | 76.86  |
> > *Inst. is an abbreviation of Instruct.
>
> > [1] Clark, P., Cowhey, I., Etzioni, O., Khot, T., Sabharwal, A., Schoenick, C., & Tafjord, O. (2018). Think you have solved question answering? try arc, the ai2 reasoning challenge. arXiv preprint arXiv:1803.05457.
> > [2] Zellers, R., Holtzman, A., Bisk, Y., Farhadi, A., & Choi, Y. (2019). Hellaswag: Can a machine really finish your sentence?. arXiv preprint arXiv:1905.07830.
> > [3] Zhou, J., Lu, T., Mishra, S., Brahma, S., Basu, S., Luan, Y., ... & Hou, L. (2023). Instruction-following evaluation for large language models. arXiv preprint arXiv:2311.07911.
> > [4] Talmor, A., Herzig, J., Lourie, N., & Berant, J. (2019, June). Commonsenseqa: A question answering challenge targeting commonsense knowledge. ACL2019, Volume 1 (Long and Short Papers) (pp. 4149-4158).

---

> ### Author Response · Authors · 2025-11-17
> **Response to Reviewer fcs2 - Part II (Question 1 and Weakness 2)**
>
> ## Question 2 and Weakness 1
> ### Have you evaluated larger LLMs (more than 7B) regarding the degradation from quantization? It might be interesting to observe whether the degradation is even smaller.
> Thanks for your insightful question, we believe your intuition is precisely correct. We have conducted additional evaluations on larger-scale models in the Qwen3 series, using identical quantization settings (bit-width, calibration data, and hyperparameters) as in our main paper.
>
> **Table R2.2.1:**
> |  | Qwen3-8B | Qwen3-8B | Qwen3-8B | Qwen3-14B | Qwen3-14B | Qwen3-14B | Qwen3-32B | Qwen3-32B | Qwen3-32B |
> | --- | --- | --- | --- | --- | --- | --- | --- | --- | --- |
> |  | Van. | AWQ | GPTQ | Van. | AWQ | GPTQ | Van. | AWQ | GPTQ |
> | **MATH** | 55.88 | 53.96 | 53.58 | 63.52 | 62.28 | 63.12 | 66.92 | 65.72 | 65.94 |
>
> We observe a clear trend: larger models exhibit stronger robustness to quantization. Accuracy degradation from full-precision to 4-bit quantized variants decreases notably with increasing parameter count, consistent with findings in recent studies [1-2]. This phenomenon arises may because larger models possess a richer and more redundant parameter space, providing greater resilience to the quantization-induced perturbations when mapping high-precision tensors into low-bit representations. Such robustness scaling with model size has also been empirically reported in recent quantization and scaling-law literature.
>
> The reason why our main paper primarily focuses on small- and medium-scale LLMs is that our main goal is to examine and recovering quantization under realistic hardware-efficiency constraints, where low-bit formats (e.g., INT4) deliver the most tangible cost–performance benefits. Therefore, we selected widely used open-source models (Qwen and LLaMA families) and quantization methods (AWQ, GPTQ, SmoothQuant) for systematic comparison.

---

> ### Author Response · Authors · 2025-11-17
> **Response to Reviewer fcs2 - Part III (Question 2)**
>
> Thank you for your insightful suggestion. I fully agree that leveraging the rich annotations in the MATH dataset to explore the relationship between quantization-induced degradation, problem difficulty, and subject category provides valuable insights. As you mentioned, the MATH dataset includes seven subject domains (algebra, geometry, number theory, combinatorics, etc.) and difficulty levels ranging from Level 1 to Level 5.
> We will incorporate a detailed correlation analysis between degradation severity and these annotations in the revised appendix, which we believe will offer both theoretical insight into quantization sensitivity across reasoning types and practical guidance for model deployment under different reasoning complexities.
>
> **Table R2.3.1:**
> |  |  | **Number Theory(%)** | **Counting &   Probability(%)** | **Intermediate   Algebra(%)** | **Algebra(%)** | **Geometry(%)** | **Prealgebra(%)** | **Precalculus(%)** |
> | --- | --- | --- | --- | --- | --- | --- | --- | --- |
> | **Qwen2.5-0.5B-Inst.** | Van. | 11.59 | 10.33 | 20.48 | 18.65 | 10.3 | 16.02 | 12.63 |
> | **Qwen2.5-0.5B-Inst.** | AWQ | 11.26 | 9.82 | 18.77 | 22.24 | 9.84 | 16.68 | 11.39 |
> | **Qwen2.5-0.5B-Inst.** | GPTQ | 11.08 | 9.8 | 18.74 | 22.57 | 9.65 | 16.71 | 11.45 |
> | **Qwen2.5-1.5B-Inst.** | Van. | 11.31 | 10.67 | 22.98 | 15.79 | 10.67 | 13.73 | 14.86 |
> | **Qwen2.5-1.5B-Inst.** | AWQ | 11.39 | 10.26 | 20.16 | 19.16 | 10.62 | 15.72 | 12.68 |
> | **Qwen2.5-1.5B-Inst.** | GPTQ | 11.57 | 10.28 | 21.74 | 18.99 | 10.28 | 13.59 | 13.56 |
> | **Qwen2.5-3B-Inst.** | Van. | 10.4 | 10.35 | 24.85 | 12.91 | 11.94 | 12.07 | 17.49 |
> | **Qwen2.5-3B-Inst.** | AWQ | 10.74 | 10.78 | 25.05 | 13.92 | 11.13 | 12.29 | 16.09 |
> | **Qwen2.5-3B-Inst.** | GPTQ | 11.23 | 9.98 | 24.49 | 14.58 | 11.2 | 12.3 | 16.22 |
> | **Qwen2.5-7B-Inst.** | Van. | 7.96 | 10.31 | 26.28 | 11.75 | 12.87 | 11.91 | 18.91 |
> | **Qwen2.5-7B-Inst.** | AWQ | 9.58 | 9.88 | 25.89 | 11.87 | 12.22 | 11.67 | 18.9 |
> | **Qwen2.5-7B-Inst.** | GPTQ | 8.56 | 10.5 | 27.13 | 11.2 | 12.39 | 11.6 | 18.62 |
>
> **Table R2.3.2:**
> |  |  | **Level 1(%)** | **Level 2(%)** | **Level 3(%)** | **Level 4(%)** | **Level 5(%)** |
> | --- | --- | --- | --- | --- | --- | --- |
> | **Qwen2.5-0.5B-Inst.** | Van. | 5.72 | 15.41 | 21.39 | 26.04 | 31.44 |
> | **Qwen2.5-0.5B-Inst.** | AWQ | 7.53 | 17.29 | 22.43 | 24.99 | 27.75 |
> | **Qwen2.5-0.5B-Inst.** | GPTQ | 7.17 | 17.35 | 22.44 | 24.99 | 28.05 |
> | **Qwen2.5-1.5B-Inst.** | Van. | 4.38 | 14.02 | 20.3 | 25.59 | 35.71 |
> | **Qwen2.5-1.5B-Inst.** | AWQ | 6.04 | 15.79 | 21.37 | 25.31 | 31.48 |
> | **Qwen2.5-1.5B-Inst.** | GPTQ | 4.41 | 14.22 | 21.03 | 26.7 | 33.65 |
> | **Qwen2.5-3B-Inst.** | Van. | 3.39 | 12.2 | 19.03 | 25.73 | 39.65 |
> | **Qwen2.5-3B-Inst.** | AWQ | 3.3 | 12.21 | 20.12 | 25.86 | 38.5 |
> | **Qwen2.5-3B-Inst.** | GPTQ | 3.16 | 12.57 | 19.42 | 26.5 | 38.35 |
> | **Qwen2.5-7B-Inst.** | Van. | 3.42 | 12.45 | 19.07 | 26.01 | 39.05 |
> | **Qwen2.5-7B-Inst.** | AWQ | 3.34 | 12.02 | 18.75 | 25.79 | 40.1 |
> | **Qwen2.5-7B-Inst.** | GPTQ | 3.38 | 12 | 19.31 | 25.19 | 40.12 |
>
> > *Van. is an abbreviation of vanilla, representing full-precision baseline.
> > *Inst. is an abbreviation of Instruct.
>
> From the subject-wise (Table 3.1), we could observe a clear and consistent trend across all model scales: quantization-induced degradation is not uniformly distributed across mathematical domains. Topics that involve multi-step symbolic manipulation—such as Intermediate Algebra, Precalculus, and Advanced Algebraic transformations—show noticeably larger performance drops under both AWQ and GPTQ, whereas domains requiring more direct recall or simpler numeric reasoning remain comparatively stable. This pattern suggests that low-bit perturbations disproportionately impact tasks requiring long-range dependency tracking and precise arithmetic transformations, aligning with our step-level error analysis in the main paper.
>
> From the difficulty-level analysis (Table 3.2), it reveals a monotonic increase in degradation with problem difficulty. While Level 1–2 questions show only marginal changes after quantization, degradation becomes more pronounced at Levels 3–5, especially for Level 5, where the gap between FP and quantized models can exceed 6–10 percentage points even for larger models.
>
> We believe these results provide strong empirical evidence that:
> - More conceptually demanding and algebraically heavy subfields of MATH are particularly vulnerable to precision reduction, and
> - Higher-difficulty problems, which require deeper reasoning are accumulate quantization noise and errors more severely.
>
> We think these findings reinforce our core claim that quantization disproportionately affects mathematical reasoning. We will include this detailed analysis in the revised appendix for completeness. Thank you again for your insightful questions.

---

### Official Review · Reviewer_51Nr · 2025-11-01

**Soundness:** 2
**Presentation:** 2
**Contribution:** 2
**Rating:** 4
**Confidence:** 4

**Summary:**

This paper presents a systematic investigation into the degradation of mathematical reasoning abilities in low-bit LLMs due to post-training quantization. The authors propose a "measure-locate-restore" loop, which involves a step-aligned error analysis to identify early-stage failures and then uses targeted "Silver Bullet" datasets with DPO to fine-tune quantized models. The study demonstrates that this approach can effectively restore mathematical reasoning performance to near full-precision baselines with minimal computational resources, while preserving the efficiency benefits of low-bit models.

**Strengths:**

1. The paper tackles the highly relevant challenge of deploying powerful LLMs in resource-constrained environments by making them efficient through quantization, while simultaneously preserving their reasoning capabilities.
2. The paper widely used PTQ methods (AWQ, GPTQ, SmoothQuant) and mathematical benchmarks (GSM8K, MATH, AIME).
3. The paper shows that significant recovery of mathematical reasoning accuracy can be achieved with as few as 332 curated examples and 3-5 minutes of GPU compute, which is a compelling result for practical deployment.

**Weaknesses:**

1. While the paper claims its framework is quantizer- and architecture-agnostic and applicable to "broader domains," the experiments are exclusively focused on mathematical reasoning tasks and a specific set of LLM families and PTQ methods. Further evidence would strengthen the broader generalizability claim.
2. The paper studies a limited variety of models, and for mathematical reasoning tasks, long CoT models are more mainstream but were not investigated. Furthermore, there was no analysis of larger models and moe models.

**Questions:**

1. In your method, are the LoRA weights also quantized in the same way during inference?
2. The study primarily focuses on mathematical reasoning. What specific modifications or considerations would be necessary to effectively apply the "measure-locate-restore" loop and "Silver Bullet" concept to other complex reasoning domains, such as logical inference or code generation?
3. The authors need to further validate the effectiveness of their method on mainstream thinking models (Qwen3), moe models (Mistral/Qwen3), and larger models (>=32B).

---

> ### Author Response · Authors · 2025-11-17
> **Response to Reviewer 51Nr - Part I (Question 1 and Question2)**
>
> Thanks for the reviewer 51Nr’s thoughtful and constructive comments. We appreciate the detailed observations, all of which directly help improve the clarity, presentation, and empirical support of our work. We hope our clarifications and newly added analyses can fully resolve the your concerns.
>
> ## Question 1 and Weakness 2
> ### In your method, are the LoRA weights also quantized in the same way during inference?
> Thank you for this question! We fully understand the importance of ensuring that the LoRA weights are also quantized, as this is essential for maintaining experimental fairness, reliability, and accurately reflecting capability changes under low-bit constraints.
>
> Yes, in our experiments, the LoRA adapters used during the recovery stage and evaluation stage operate under the same quantization regime as the host model. Specifically, we adopt QLoRA for 4-bit quantized models (see Sections 3.1.2 and 3.2.3), ensuring that the entire inference stack—including LoRA weights—remains within the same low-bit arithmetic precision. This design preserves end-to-end quantization consistency and prevents any hidden precision leakage that could otherwise bias the recovery performance.
> ## Question 2 and Weakness 1
> ### The study primarily focuses on mathematical reasoning. What specific modifications or considerations would be necessary to effectively apply the "measure-locate-restore" loop and "Silver Bullet" concept to other complex reasoning domains, such as logical inference or code generation?
> Thank you for raising this valuable question! This framework is designed to generalize naturally to any step-structured reasoning domain. Just as you mentioned, as its core principle, the "measure–locate–restore" loop operates independently of domain-specific semantics. The key requirement is that the reasoning process can be decomposed into interpretable intermediate steps, which allows systematic error localization and targeted capability restoration.
>
> In fact, in our ongoing extension to coding tasks, we largely retain the same overall workflow used for mathematical reasoning, and also with a few domain-specific adjustments have been list below:
> - Simplified error measurement and localization. Unlike mathematical problems, where we rely on an automated chain-of-thought error, analysis pipeline involving a pivot model and majority voting across multiple LLM judges to ensure consistency. Code generation tasks provide a more direct and objective evaluation signal. By executing the generated code in a sandbox environment, we can automatically categorize errors via compiler feedback (mainly are syntax errors, runtime errors, logical test failures) with a clearly locatation (which line), thereby simplifying the "measure" and "locate" stages.
> - Error hierarchy refinement. In mathematics, we treat the first erroneous reasoning step as the most critical, since subsequent derivations usually depend on it. In contrast, we find code errors sometimes are not always dominated by the first failure point. Multiple early issues may coexist, and their relative importance varies depending on functional dependencies. Hence, we adapt the "locate" stage to trace and weight multiple error categories rather than identifying only the initial error.
>
> We believe these adaptations demonstrate that our measure–locate–restore paradigm and "Silver Bullet" methodology can be flexibly transferred to other structured reasoning domains.

---

> ### Author Response · Authors · 2025-11-17
> **Response to Reviewer 51Nr - Part II (Question 3)**
>
> ## Question 3
> ### The authors need to further validate the effectiveness of their method on mainstream thinking models (Qwen3), moe models (Mistral/Qwen3), and larger models (>=32B).
> Thank you for this insightful suggestion. We fully agree that recent advances, especially Chain-of-Thought (CoT) and Mixture-of-Experts (MoE) models have drawn growing attention within the LLM community due to their superior performance on complex reasoning tasks.
>
> Regarding our experimental focus, as clarified in the Introduction (Section 1), this work primarily targets **resource-constrained deployment scenarios** and to recovering the quantized models' reasoning ability, where large models (≥32B) are impractical due to their high memory and computation overhead. Similarly, CoT-style "thinking" models often trade computational efficiency for longer reasoning chains, leading to substantially higher average output tokens and total response time. Our goal is **to investigate quantization under realistic hardware-efficiency constraints, focusing on small- to medium-scale models where low-bit integer formats (e.g., INT4) bring the most tangible benefits**. So we chose the most popular small and medium scale open-source model families(Qwen and Llama) and most popular quantization methods (AWQ and GPTQ) for our experiments.
>
> To strengthen the generality of our conclusions, we have conducted additional experiments on the Qwen3 series, which natively support a "thinking mode" as the default configuration. These models already integrate internal CoT-like mechanisms, making them representative of both standard and "thinking" (CoT) LLMs. We evaluated Qwen3-8B, Qwen3-14B, and Qwen3-32B under identical quantization settings (bit-width, calibration set, and hyperparameters) as in our main paper.
>
> **Table R1:**
> |  | Qwen3-8B | Qwen3-8B | Qwen3-8B | Qwen3-14B | Qwen3-14B | Qwen3-14B | Qwen3-32B | Qwen3-32B | Qwen3-32B |
> | --- | --- | --- | --- | --- | --- | --- | --- | --- | --- |
> |  | Van. | AWQ | GPTQ | Van. | AWQ | GPTQ | Van. | AWQ | GPTQ |
> | **MATH** | 55.88 | 53.96 | 53.58 | 63.52 | 62.28 | 63.12 | 66.92 | 65.72 | 65.94 |
>
> > *Van. is an abbreviation of vanilla, representing full-precision baseline.
>
> We believe these results yield several key insights:
> - **Larger models show greater quantization robustness**. Accuracy degradation from full-precision to quantized versions diminishes significantly as model size increases, consistent with observations from other studies. We believe because larger models possess a richer parameter space, which grants stronger robustness to quantization-induced numerical errors when mapping from high-precision to low-precision data formats [1–2].
> - **Mild regularization effects appear in quantized models**. You can find an interesting result that Qwen3-14B under GPTQ slightly outperforms its vanilla counterpart, suggesting that moderate compression may enhance generalization, a phenomenon also noted in recent quantization studies. Notably, this effect has also been frequently cited as one of the underlying reasons why Quantization-Aware Training (QAT) can sometimes improve the generalization ability of post-quantized models [3].
> - **Scalability of the "Silver Bullet" principle**. The consistently small degradation across larger models supports our hypothesis that targeted recovery using compact, well-curated data is even more effective for high-capacity models with stronger learning abilities.
>
> We will also consider validating the effectiveness of "Silver Bullet" on smaller-scale MoE models in future work (although, at present, I think there are still no sufficiently compact open-sourced MoE models suitable for on-device deployment). From a hardware perspective, the sparse activation pattern inherent to MoE architectures makes them highly promising for future computation–memory co-optimized deployment under resource-constrained environments.
> > [1] Lee, J., Park, S., Kwon, J., Oh, J., & Kwon, Y. (2024). Exploring the Trade-Offs: Quantization Methods, Task Difficulty, and Model Size in Large Language Models From Edge to Giant. arXiv preprint arXiv:2409.11055.
> > [2] Shen, A., Lai, Z., & Li, D. (2024, September). Exploring quantization techniques for large-scale language models: Methods, challenges and future directions. In Proceedings of the 2024 9th International Conference on Cyber Security and Information Engineering (pp. 783-790).
> > [3] AskariHemmat, M., Jeddi, A., Hemmat, R. A., Lazarevich, I., Hoffman, A., Sah, S., ... & David, J. P. (2024). Qgen: On the ability to generalize in quantization aware training. arXiv preprint arXiv:2404.11769.

---

### Official Review · Reviewer_xpx2 · 2025-11-04

**Soundness:** 3
**Presentation:** 3
**Contribution:** 3
**Rating:** 6
**Confidence:** 4

**Summary:**

This paper presents a systematic investigation into the degradation of mathematical reasoning in low-bit quantized LLMs. The authors introduce a "measure-locate-restore" pipeline to first diagnose and then mitigate this performance loss. Through an error analysis across various models, quantization methods, and benchmarks, they find that post-training quantization predominantly introduces early-stage method and execution errors rather than high-level conceptual mistakes. Based on this diagnosis, they propose a lightweight mitigation strategy. They automatically construct compact, targeted "Silver Bullet" datasets of failure cases and use DPO to fine-tune the quantized models. Their experiments demonstrate that this approach can recover reasoning performance to near full-precision levels with minimal data (as few as 332 examples) and compute (3-5 minutes on a single GPU), without compromising the efficiency benefits of quantization.

**Strengths:**

1. The paper is well-written and easy to follow.
2. The work addresses a highly practical problem. The deployment of LLMs on resource-constrained hardware is a significant bottleneck, and understanding and rectifying performance degradation from essential compression techniques like quantization is of great importance to the field.
3. A  key contribution is the detailed, step-aligned error analysis. The finding that quantization disproportionately increase method errors and execution errors, and that these errors manifest early in the reasoning chain.
4. The proposed method demonstrates ability to restore performance with a very small number of examples of fine-tuning.

**Weaknesses:**

1. The definitions for the four high-level error types (Conceptual, Method, Execution, Reasoning) appear to have some overlap, which could lead to subjective classification. For example, misusing a formula in an unsuitable context could be interpreted as a Conceptual Error (misunderstanding the problem's constraints) or a Method Error (choosing an inappropriate method).

**Questions:**

1. In your discussion of "Why Reasoning Errors seem to vanish" (line 412), you explain that earlier procedural errors mask subsequent logical flaws. Does this imply that after your restoration process fixes these initial errors, "Reasoning Errors" might re-emerge as the next performance bottleneck in the now-stronger models?
2. The paper highlights that quantization primarily increases "Method" and "Execution" errors. Your ablation study (Table 2) shows that fine-tuning on Conceptual Error also yields the very strong results. This seems slightly counter-intuitive; Do you have an intuition for why this is the case?

---

> ### Author Response · Authors · 2025-11-17
> **Response to Reviewer xpx2 - Part I**
>
> ## Question1
> ### In your discussion of "Why Reasoning Errors seem to vanish" (line 412), you explain that earlier procedural errors mask subsequent logical flaws. Does this imply that after your restoration process fixes these initial errors, "Reasoning Errors" might re-emerge as the next performance bottleneck in the now-stronger models?
>
> Thank you for the question, I would like to offer a brief clarification to address your concern.
>
> For the purpose of enabling a more precise quantitative analysis, our pipeline is designed to record only the first error that appears in the reasoning trace. Under this protocal setting, we observed the somewhat counter-intuitive phenomenon that the proportion of Reasoning Errors actually decreases after quantization.
> Conceptually, this indeed suggests the picture that the reviewer points out: if we apply our "measure–locate–restore" loop to reduce early procedural and execution failures, then higher-level Reasoning Errors are likely to re-emerge as the dominant residual bottleneck in a stronger model. This is analogous to human learners: once basic arithmetic and procedural routines are stabilised, deeper logical reasoning becomes the main source of mistakes.
>
> In the current work, we did not re-run the full large-scale error taxonomy over all models after capability restoration. However, in a limited manual inspection of post-restoration failures, we did observe more instances that would be classified as reasoning-level gaps rather than simple arithmetic slips. We will clarify in the final version that:  The "vanishing reasoning errors" phenomenon is specific to the first-error protocol under quantization.  A natural direction for future work is to perform multi-round restoration, where the first round focuses on Execution/Method failures and subsequent rounds explicitly target the remaining Reasoning Errors.
>
> We will add this clarification to the Discussion section to avoid over-interpreting the current error distribution
>
> ## Question 2
> ### The paper highlights that quantization primarily increases "Method" and "Execution" errors. Your ablation study (Table 2) shows that fine-tuning on Conceptual Error also yields the very strong results. This seems slightly counter-intuitive; Do you have an intuition for why this is the case?
>
> Thank you for this thoughtful question. Let us first connect this to our discussion around Figure 2. As we clarified elsewhere, we believe the decrease in the measured share of Reasoning Errors for quantized models does NOT mean that quantization improves high-level reasoning. It is a measurement artefact of our evaluation protocol, which always records only the first erroneous step in each solution trace. Quantization amplifies early-stage numerical and procedural mistakes, so Execution/Method errors appear earlier in the chain and "mask"(or "hide") later logical flaws; this shifts the observed error distribution without changing the underlying difficulty of reasoning.
>
> So we believe the strong performance of Conceptual-Error-only (CE-only) training is not contradictory, for two reasons:
> 1. Labels are based on the first error, not the only error. A "Conceptual Error" label simply means that the first detected failure in the trace is conceptual. The same instance often contains downstream method and execution mistakes as consequences of the wrong setup. In our Silver Bullet construction, we truncate the quantized trace at that first erroneous step and ask a strong baseline model to regenerate the rest of the solution, producing a positive trace that contains the entire corrected chain, including appropriate method choices and accurate computations. Thus, CE-only training does not exclusively supervise conceptual knowledge, it also implicitly teaches correct procedural and computational behavior on hard and conceptually tricky cases.
> 2. Conceptual cases provide high-leverage upstream supervision. Conceptual failures tend to occur early in the solution, we have done a lot of case studies and find that because misinterpreting the problem statement, variable roles, or constraints derails all subsequent steps. Aligning the model’s conceptual understanding and problem setup therefore has a cascading effect: once the problem is framed correctly, many downstream method/execution mistakes disappear or become much easier to avoid.
> To summarize, our results suggest that quantization predominantly manifests as Method/Execution errors in our taxonomy, while high-impact conceptual counterexamples are particularly effective training signals for restoring overall robustness.
>
> We thank your good question again and we will add this intuition and a short clarification near Figure 2 and Table 2 in the revised version.

---

### Author Response · Authors · 2025-11-27
**Updated Manuscript and Highlighted Revisions**

We thank all reviewers for their constructive comments. We have uploaded a revised version that clarifies Figure 2 and the error taxonomy, explains our quantization and model choices, and adds supporting experiments. We have highlighted all changes in blue for easier reference.

Two questions emerge consistently across the reviews. While these points have been clarified in the revised version, we provide a unified explanation here for completeness:

### Q1. Error type shift after quantization

Our error assessment pipeline assigns each failing trajectory a single label given by the first erroneous step. Reasoning type failures usually appear later in the solution, while conceptual and execution mistakes often occur earlier. Quantization makes these earlier mistakes more frequent, so they more often become the first error and receive the label. This first error protocol reduces the measured share of Reasoning Errors even when later reasoning flaws are still present. In the revised manuscript, Section 4.2 and Appendix D describe this protocol explicitly and clarify that "mask" or "hide" refers only to this **statistical** effect.

### Q2. Quantization and model choices

Our aim is to understand how widely used low bit deployment recipes affect mathematical reasoning and to mitigate this degradation under realistic resource constraints.

**Quantization settings.** We focus on four bit post training weight quantization with AWQ and GPTQ and eight bit weight activation quantization with SmoothQuant because these are standard PTQ recipes with broad hardware support. Our measure locate restore pipeline is agnostic to bit width and can be applied once stable implementations are available.

**Model settings.** Larger models are empirically more robust to quantization, so the accuracy gap between full precision and four bit variants shrinks as scale increases. This makes very large models less informative for analyzing severe quantization induced degradation, while edge and device deployments typically rely on models in the sub seven billion parameter range. We therefore focus on Qwen and Llama models from zero point five to seven billion parameters. The revised Appendix E additionally reports experiments on larger Qwen3 models under the same quantization settings; the results follow the same robustness trend and do not change our conclusions.

---

### Author Response · Authors · 2025-12-04
**Overarching Response**

Dear Area Chair,

As the rebuttal period concludes, we appreciate the constructive discussion and the concerns raised and we briefly summarize the main contributions and strengths of this work.

**Our paper presents an ability aware framework that connects measurement, attribution and targeted restoration of quantization induced degradation in mathematical reasoning for LLMs.** By systematically evaluating mainstream PTQ methods (AWQ, GPTQ, SmoothQuant) on leading open source model families such as LLaMA 3 and Qwen 2.5 / Qwen 3 under realistic 4 bit and 8 bit settings, we show that low bit quantization can reduce math reasoning accuracy by up to 69.81 percent, with smaller models and harder tasks being most affected. Building on this, we introduce a step aligned error taxonomy and an automatic assessment pipeline that classifies failures into conceptual, method, execution and reasoning errors with high agreement to human annotators, and reveals that quantization mainly harms procedural and execution skills rather than general knowledge.

On top of this analysis, we propose a lightweight yet effective recovery pipeline that constructs a compact "Silver Bullet" dataset by localizing the first erroneous step, truncating the failing trace and asking a stronger model to complete it. Fine tuning the quantized models directly in the low bit regime on only a few hundred such stepwise contrastive pairs and a few minutes on a single GPU restores mathematical reasoning performance to the level of, or very close to, the full precision baseline, while leaving broader capabilities such as MMLU essentially unchanged and preserving the efficiency advantages of 4 bit deployment. The pipeline is model and quantization method agnostic, depends only on post quantization behavior, and our preliminary results indicate that the same recipe can extend to other step structured domains such as code, which makes the work practically useful for real world low bit deployment of reasoning models.

For the questions raised by the reviewers, we have provided detailed responses and updated the revised PDF accordingly. All modifications are highlighted in blue to help you navigate the changes quickly. We sincerely thank you again for your time and constructive efforts during the review process.

Best regards,
Team 17461

---

### Meta-Review · Area_Chair_DrWM · 2026-01-13

**Summary:**

Reviewers broadly agreed that the paper addresses an important and practical problem—the degradation of mathematical reasoning under low-bit post-training quantization—and found the empirical analysis and mitigation pipeline promising. However, several concerns informed the decision discussion:

- Generality and scope: Multiple reviewers questioned whether conclusions drawn from mathematical reasoning and small-to-medium models (≤7B in the main paper) generalize to other reasoning domains, larger models, MoE models, or more aggressive quantization (2–3 bit).

- Clarity and rigor of the error taxonomy and ablations: Some reviewers noted potential overlap between error categories and initially found it unclear whether the “error-aware” Silver Bullet selection provided clear advantages over random or generic supervision.

- Experimental completeness and fairness: Concerns were raised about missing details in the fine-tuning protocol (format alignment, LoRA/QLoRA consistency), and whether improvements might stem from additional math exposure rather than targeted quantization repair.

- Positioning relative to prior work: A minority of reviewers questioned whether focusing on 4-bit PTQ and small data regimes limited the paper’s novelty or research ambition compared to work pushing ultra-low-bit quantization.

Overall, while strengths (clear empirical trends, practical mitigation with very small compute/data, and careful rebuttal) were widely acknowledged, lingering doubts about scope and breadth led to some borderline or below-threshold scores.

**Reviewer Concerns:**

Substantially addressed by the rebuttal and revision:

- Error taxonomy clarity and “vanishing reasoning errors”: The authors convincingly clarified the first-error protocol and explained why reasoning errors appear to decrease after quantization, resolving confusion raised by multiple reviewers.

- Effectiveness of error-aware supervision: Additional explanation and controlled comparisons strengthened the case that Silver Bullet selection yields more sample-efficient gains than random or non-step baselines under fixed budgets.

- Experimental transparency and fairness: Detailed clarifications on format-alignment training, adapter usage (LoRA vs. QLoRA), quantization consistency, and added ablations reduced concerns about unfair comparisons or hidden precision leakage.

- Generality beyond math (to some extent): Added results on non-math benchmarks (ARC, HellaSwag, CommonsenseQA, IFEval) supported the claim that math reasoning is disproportionately affected by quantization.

Partially addressed or still outstanding:

- Ultra-low-bit (2–3 bit) quantization: While the authors provided strong practical and empirical justification for excluding 2–3 bit PTQ (instability, corrupted outputs, lack of reproducible baselines), some reviewers still viewed this as a missed research opportunity rather than a flaw.

- Broader model classes (MoE, very large models): Additional Qwen3 experiments mitigated this concern, but MoE models remain untested, and some reviewers may still see the scope as conservative.

- Perceived niche of “low-data regime”: The rebuttal clarified that “low-data” refers to the recovery phase rather than the math domain itself, but at least one reviewer remained unconvinced about how broadly this framing applies.

**Reviewer Scores:**

Based on the tone of the rebuttal responses and follow-up comments, I estimate the following likely score changes had reviewers fully participated in discussion:

Reviewer xpx2 (initial rating: 6): Likely unchanged. Their main questions were directly answered, but their initial stance was already cautiously positive.

Reviewer 51Nr (initial rating: 4): Likely maintain. Added large-model experiments, clarification on LoRA quantization, and domain generalization likely alleviated most core concerns.

Reviewer fcs2 (initial rating: 6): Likely maintain. Additional benchmarks and larger-model analyses directly addressed their weaknesses.

Reviewer bsjF (initial rating: 4): Likely maintain. Many technical and clarity issues were addressed, but philosophical concerns about scope (2–3 bit, low-data framing) appear to persist.

---

### Decision · Program_Chairs · 2026-01-26

Reject